# Characterisation of cold-selective lamina I spinal projection neurons in the mouse

Aimi N Razlan[1†], Wenhui Ma[1†], Allen C Dickie[1†], Erika Polgar[1†], Anna G McFarlane[1], Mansi Yadav[1], Andrew H Cooper[1], Douglas Strathdee[2], Masahiko Watanabe[3], Andrew M Bell[1], Andrew J Todd[1*], Junichi Hachisuka[1*]

[1]School of Psychology and Neuroscience, College of Medical, Veterinary and Life Sciences, University of Glasgow, Glasgow, United Kingdom; [2]School of Cancer Sciences, College of Medical, Veterinary and Life Sciences, University of Glasgow, Bearsden, United Kingdom; [3]Department of Anatomy, Hokkaido University School of Medicine, Sapporo, Japan

**\*For correspondence:**
Andrew.Todd@glasgow.ac.uk
(AJT);
Junichi.Hachisuka@glasgow.ac.
uk (JH)

[†]These authors contributed
equally to this work

**Competing interest:** The authors
declare that no competing
interests exist.

**Reviewing Editor:** John R
Huguenard, Stanford University
School of Medicine, United
States

## eLife Assessment

This **important** study offers insights into the anatomical and physiological features of cold-selective lamina I spinal projection neurons. The evidence supporting the authors' claims is **convincing**, although including a larger sample size and more quantification would have strengthened the study, and the claims of monosynaptic connectivity would benefit from further experimental evidence. The work will interest those in the field of somatosensory biology, especially researchers studying spinal cord dorsal horn circuits and projection neuron cell types

**Abstract** Skin cooling is detected by primary afferents that express the Trpm8 channel, but how this information is conveyed to the brain remains poorly understood. We have previously identified a population of lamina I projection neurons belonging to the anterolateral system (ALS) that receive numerous contacts from Trpm8-expressing primary afferents. Here, using a semi-intact somatosensory preparation, we provide evidence that these cells correspond to the cold-selective ALS neurons identified in previous physiological studies. We also confirm the presence of synapses from Trpm8 afferents onto these cells at the ultrastructural level and with optogenetics. Based on our previous transcriptomic findings, we identify calbindin as a molecular marker, and show that this can be used to target the cold-selective ALS neurons for anterograde tracing studies. We provide evidence that they project to brain regions that have been implicated in thermosensation: the rostralmost part of the lateral parabrachial area, the caudal part of the periaqueductal grey matter, and the posterior triangular and ventral posterolateral nuclei of the thalamus. Our findings provide important insights into the organisation of neuronal circuits that underlie thermoregulation and the perception of cold stimuli applied to the skin.

## Introduction

The anterolateral system (ALS) consists of spinal cord neurons that project to various brain regions, including the caudal ventrolateral medulla (CVLM), the nucleus of the solitary tract (NTS), the lateral parabrachial area (LPB), the periaqueductal grey matter (PAG), the superior colliculus and the thalamus. ALS neurons are required for the perception of pain, itch and skin temperature, as shown by the loss of these sensations on the side contralateral to an anterolateral cordotomy (*Javed et al., 2020*; *Lahuerta et al., 1990*). Although axons belonging to this pathway ascend in the anterolateral quadrant in humans, they are located in the dorsolateral white matter in rodents (*Chen et al., 2024*;

*Ma et al., 2025a*; *Ma et al., 2025b*; *McMahon and Wall, 1983*). ALS neurons account for only a small proportion (probably less than 1%) of spinal cord neurons (*Chung et al., 1984*; *Abraira and Ginty, 2013*) and are unevenly distributed, with a relatively high density in lamina I and the lateral spinal nucleus (LSN), and scattered cells in deeper laminae (III-VII) and around the central canal (*Ma, 2022*; *Todd, 2010*; *Wang et al., 2022*; *Wercberger and Basbaum, 2019*). ALS neurons are functionally heterogeneous, and we recently identified five distinct transcriptomic populations, which we named ALS1-5 (*Bell et al., 2024*). This was based on single-nucleus RNA sequencing of a major subset of these cells, those that transiently express the transcription factor Phox2a during development (*Roome et al., 2020*).

Perception of skin cooling is dependent on primary afferent neurons that express the Trpm8 channel (*Peier et al., 2002*; *McKemy et al., 2002*; *Lewis and Griffith, 2022*; *Knowlton et al., 2013*; *Pogorzala et al., 2013*), and these terminate mainly in lamina I of the spinal cord (*Takashima et al., 2007*; *Kim et al., 2014*; *Dhaka et al., 2008*). Consistent with this, it has been shown that many lamina I projection neurons respond to cool or cold stimuli applied to the skin, with a distinctive subset of 'cold-selective' neurons responding predominantly or exclusively to skin cooling (*Craig and Kniffki, 1985*; *Craig et al., 2001*; *Craig and Dostrovsky, 2001*; *Craig and Hunsley, 1991*; *Dostrovsky and Craig, 1996*; *Zhang et al., 2006*; *Allard, 2019*; *Andrew, 2009*; *Hachisuka et al., 2016*; *Hachisuka et al., 2020*; *Chisholm et al., 2021*).

How information is transmitted from Trpm8-expressing primary afferent neurons to these cold-selective lamina I ALS cells is currently not known, although a recent report has suggested that this occurs via a dedicated population of excitatory interneurons (*Lee et al., 2025*). However, we had previously identified a population of lamina I ALS neurons that received dense synaptic input from Trpm8-expressing primary afferents (*Bell et al., 2024*). These neurons had dendrites that were intimately associated with bundles of Trpm8-positive afferents, while their cell bodies generally also received numerous contacts, and were sometimes surrounded by these afferents. We, therefore, proposed that these cells correspond to the cold-selective neurons described above, and also provided evidence that they belonged to the ALS3 population (*Bell et al., 2024*).

The aims of this study were to characterise and validate the *Trpm8*Flp mouse line that we had used to reveal this input, to confirm the presence of synaptic input, and to test the prediction that the

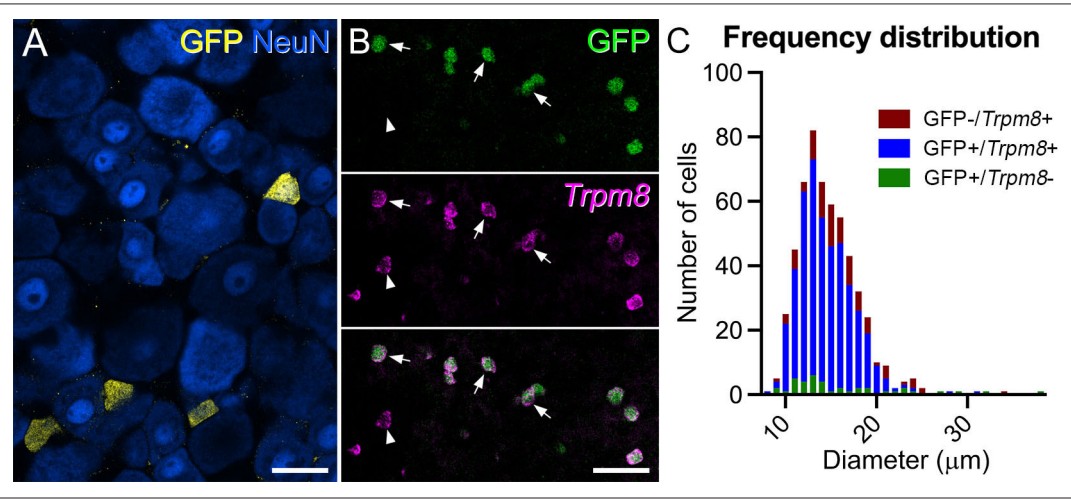

**Figure 1.** The *Trpm8*Flp mouse line captures Trpm8-expressing primary afferent neurons. (**A**) Immunohistochemical staining for GFP (yellow) and NeuN (blue) in a dorsal root ganglion from a *Trpm8*Flp;RCE:FRT mouse. GFP is present in some of the small neurons. (**B**) Fluorescence in situ hybridisation with a probe for *Trpm8* mRNA (magenta) superimposed on GFP native fluorescence (green). The great majority of GFP cells contain *Trpm8* message, and *vice versa*. Some of the double-labelled cells are indicated with arrows, while the arrowhead points to a *Trpm8*-positive cell that lacks GFP. (**C**) Frequency distribution of soma diameter for cells that were GFP-/*Trpm8*+ (red bars), GFP+/*Trpm8*+ (blue bars), or GFP+/*Trpm8*- (green bars). Scale bars: (**A**) = 25 μm, (**B**) = 50 μm.

The online version of this article includes the following figure supplement(s) for figure 1:

**Figure supplement 1.** Immunohistochemical characterisation of GFP-expressing somata in dorsal root ganglia of *Trpm8*Flp;RCE:FRT mice.

lamina I projection neurons with dense synaptic input from Trpm8-expressing primary afferents do indeed correspond to the cold-selective population described in physiological studies (*Craig and Kniffki, 1985*; *Craig et al., 2001*; *Craig and Dostrovsky, 2001*; *Craig and Hunsley, 1991*; *Dostrovsky and Craig, 1996*; *Zhang et al., 2006*; *Allard, 2019*; *Andrew, 2009*; *Hachisuka et al., 2016*; *Hachisuka et al., 2020*; *Chisholm et al., 2021*). In addition, since Calb1 (which encodes the calcium-binding protein calbindin) is one of the top differentially expressed genes in the ALS3 population, we used mice in which Cre recombinase is targeted to the Calb1 locus (*Calb1*^Cre) to investigate the projections of the ALS3 cells to the brain.

## Results

### Validation of *Trpm8*^Flp mouse line and characterisation of Flp-expressing cells

*Trpm8*^Flp mice were crossed with the Flp reporter line RCE:FRT to generate *Trpm8*^Flp;RCE:FRT mice, in which cells that express Flp at any stage during development are labelled with GFP. Tissue was obtained from mice aged between 5–9 weeks for this part of the study. We used a stereological technique to quantify neurons in an L4 dorsal root ganglion (DRG) from three of these mice and found that 5.7% (range 5.4–6.7%) of all DRG neurons contained GFP (*Figure 1A*). We then performed fluorescence in situ hybridisation (FISH) to examine the extent of overlap between GFP and *Trpm8* mRNA (*Figure 1B*). In DRGs from 3 mice, we found that 92% (88–98%) of GFP-containing cells had *Trpm8* message, while 83% (80–86%) of cells with *Trpm8* were GFP-positive. These findings show a high degree of fidelity in the *Trpm8*^Flp line, and are consistent with the reported expression of *Trpm8* mRNA in 5–10% of mouse DRG cells (*Peier et al., 2002*). We also used this tissue to determine the soma size of the GFP-positive cells (*Figure 1C*), and found that these were small, with a mean diameter of 14.6 μm (±3.4 μm, SD), showing a very similar size range to that reported for cells captured with a different *Trpm8*^Flp mouse line (*Qi et al., 2024*). Trpm8-positive cells that lacked GFP, as well as the few GFP-labelled cells that lacked *Trpm8* mRNA, showed a similar size distribution.

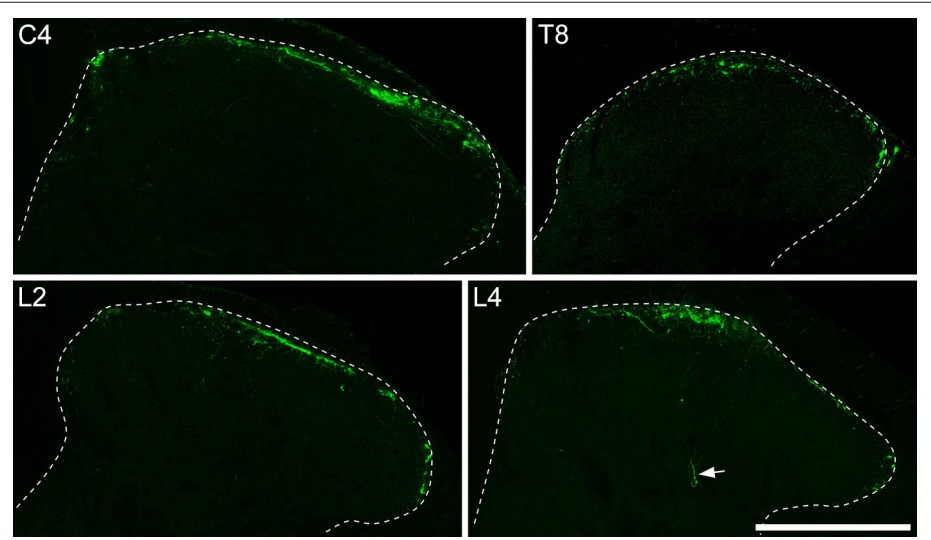

**Figure 2.** Distribution of Trpm8 afferents in the spinal dorsal horn. Immunohistochemical staining for GFP at different segmental levels through the spinal cord in *Trpm8*^Flp;RCE:FRT mice. GFP-labelled axons are largely restricted to lamina I, although occasional fibres penetrate into deeper parts of the dorsal horn (arrow). Within lamina I, the distribution of GFP axons is patchy, and does not occupy the entire mediolateral extent. Scale bar = 200 μm.

The online version of this article includes the following figure supplement(s) for figure 2:

**Figure supplement 1.** Expression of Vglut3 and lack of detectable Trpv1 in the central terminals of Trpm8 afferents.

Transcriptomic studies have demonstrated that Trpm8-expressing primary afferents can co-express certain genes that serve as markers for primary afferent populations, such as *Slc17a8* (which encodes Vglut3), *Trpv1,* and *Tac1* (the gene coding for substance P), but lack other neuropeptides, such as calcitonin gene-related peptide (CGRP) and somatostatin (*Sharma et al., 2020*; *Zeisel et al., 2018*). We therefore performed immunohistochemistry to look for expression of the corresponding proteins/peptides in dorsal root ganglia from three mice in each case. We found that 11.9% (9.8–13.8%) of GFP-containing cells were immunoreactive for Vglut3 and 26.3% (21.9–30.9%) for Trpv1 (*Figure 1—figure supplement 1A–D*). Among the neuropeptides, we found minimal co-localisation with CGRP (mean 2.8%, range 1.6–4.5%) and none with somatostatin. However, 56% (range 55.8–56.3%) of the GFP cells contained substance P, although this was expressed at relatively low levels (*Figure 1—figure supplement 1G–M*). These results are, therefore, consistent with the transcriptomic findings described above (*Sharma et al., 2020*; *Zeisel et al., 2018*).

Central projections of Trpm8-expressing primary afferents have been shown to terminate mainly in lamina I[20-22], and we found a similar pattern in the spinal cords of *Trpm8*[Flp];RCE:FRT mice, with the great majority of GFP-labelled axons remaining in lamina I, and very few passing ventrally (*Figure 2*). A similar pattern of labelling was observed in all spinal segments examined. We had previously noted that when lamina I was viewed in horizontal sections, Trpm8 afferents were arranged in interweaving bundles that frequently followed the dendrites of a distinctive subset of projection neurons (*Bell et al., 2024*). Consistent with this, we found that Trpm8 afferents labelled with GFP showed a discontinuous arrangement within lamina I when viewed in transverse sections and did not occupy the whole medio-lateral extent of the lamina (*Figure 2*). The GFP-labelled afferents did not show any obvious restriction in the medio-lateral axis, and could be seen in medial, central, and lateral parts of lamina I. Since the great majority of Trpm8-expressing afferents contain GFP in this genetic cross, it is unlikely that the large spaces between bundles of GFP-labelled afferents in lamina I are filled by Trpm8-expressing afferents that lack GFP. This 'incomplete' innervation of lamina I by Trpm8 afferents is therefore likely to be genuine.

It has been reported that some lamina I projection neurons were associated with boutons that were immunoreactive for Vglut3 (*Li et al., 2015*) and since 12% of Trpm8 afferent cell bodies contained Vglut3, we looked for expression of the transporter in their central terminals. In transverse sections, Vglut3-immunoreactive axons formed a dense band in inner lamina II that corresponds to central terminals of C-low threshold mechanoreceptors (C-LTMRs) (*Seal et al., 2009*; *Dickie et al., 2019*). However, there were a few Vglut3-immunoreactive boutons in lamina I that were also GFP-positive, indicating that central terminals of some Trpm8-expressing primary afferents had detectable levels of the transporter (*Figure 2—figure supplement 1A–F*). We generated a mouse genetic cross (*Phox-2a*::Cre;Ai9;*Trpm8*[Flp];RCE:FRT) in which a subset of ALS cells express tdTomato and Trpm8 afferents express GFP. In horizontal sections from these mice, we identified lamina I projection neurons that were associated with numerous GFP-labelled (Trpm8-expressing) boutons and found that some of these boutons contained Vglut3 (*Figure 2—figure supplement 1G–J*). The proportion of GFP-labelled boutons that contained Vglut3 varied considerably between cells. Since C-LTMRs do not appear to arborise in lamina I (*Qi et al., 2024*), it is likely that the Vglut3-containing boutons that have been found to contact projection neurons in this lamina (*Li et al., 2015*) belong to Trpm8 afferents. Even though many Trpm8-positive cell bodies showed Trpv1 and substance P immunoreactivities, we were not able to detect either of these in their central terminals (*Figure 2—figure supplement 1K–M*).

Although we did not investigate the distribution of GFP in the brains of *Trpm8*[Flp];RCE:FRT mice in detail, we noted that there was dense axonal labelling in the spinal trigeminal nucleus, while scattered GFP-labelled cell bodies were present in various brain regions that are known to contain Trpm8-expressing neurons, including the preoptic area and the reticular nucleus of the thalamus (*Ordás et al., 2021*) (data not shown).

## Synaptic input from Trpm8-expressing afferents to lamina I projection neurons

We had previously shown that ~20% of lamina I projection neurons retrogradely labelled from the LPB received numerous contacts from GFP-labelled afferents in *Trpm8*[Flp];RCE:FRT mice, and that these contacts were associated with punctate staining for Homer1 (*Bell et al., 2024*), a marker for glutamatergic synapses (*Gutierrez-Mecinas et al., 2016*). To reveal synapses formed between Trpm8 afferents

and these projection neurons at the ultrastructural level, we used a combined confocal/electron microscopic method (*Todd, 1997*) on spinal cord tissue from *Phox2a*::Cre;Ai9;*Trpm8*^Flp;RCE:FRT mice. This genetic cross was chosen because we had previously shown that the Phox2a-lineage includes many lamina I ALS neurons with dense input from Trpm8 afferents (*Bell et al., 2024*). Horizontal sections from lumbar spinal cord segments of 2 of these mice were immunostained to reveal tdTomato and GFP, and 3 tdTomato-labelled cells with numerous contacts from GFP-labelled axons were scanned with a confocal microscope (*Figure 3*, *Figure 3—figure supplements 1 and 2*). The tissue was subsequently processed with an immunoperoxidase method to label the GFP-positive axons with diaminobenzidine (DAB), and examined with an electron microscope (*Figure 3*, *Figure 3—figure supplement 2*). Although the cell bodies and dendrites of the projection neurons did not contain an electron-dense label, they could be readily identified by their location within the ultrathin sections, by their association with numerous DAB-labelled axonal boutons and by the morphology of their somata and proximal dendrites (*Figure 3A and B*). Because of the very high density of GFP-labelled boutons that were seen to contact the cells in the confocal images, it was not always possible to relate individual boutons from confocal images to those seen with the electron microscope. However, we were able to identify numerous axosomatic and axodendritic synapses (a minimum of 10 synapses per cell) between the GFP-immunoreactive boutons and the lamina I projection neurons. These could be recognised by the presence of darkening of the membranes and, in some cases, a faint postsynaptic density, together with clustering of synaptic vesicles on the presynaptic side (*Figure 3B-D*, *Figure 3—figure supplement 2C-J*). Interestingly, the synapses between the GFP-containing boutons and the projection neurons generally had relatively faint postsynaptic densities and only small regions where the vesicles were clustered on the presynaptic side. These findings, together with our previous observation of Homer1-immunoreactive puncta located between GFP-expressing afferent boutons and the cell bodies and dendrites of retrogradely labelled lamina I projection neurons in *Trpm8*^Flp;RCE:FRT mice (*Bell et al., 2024*), indicate that ALS cells belonging to this population receive numerous synapses from Trpm8-expressing primary afferents.

To provide electrophysiological evidence for synapses between Trpm8 afferents and this population of lamina I ALS cells, we carried out optogenetic experiments. Two *Trpm8*^Flp;RCE:FRT mice received intraperitoneal injections of AAV.PHP.S.Flp^ON.ChR2_YFP as neonates to label Trpm8-expressing primary afferents with channelrhodopsin. They were then injected with AAV11.tdTomato into the CVLM or LPB ~5 weeks later (*Figure 3E*). Whole-cell patch-clamp recordings were subsequently obtained from tdTomato-labelled lamina I neurons in a whole-cord preparation. We used the *Trpm8*^Flp;RCE:FRT cross for these experiments in order to allow unbiased identification of ALS neurons that were surrounded by Trpm8 afferents. Importantly, although the viral delivery approach that we used will have specifically targeted Trpm8 afferents, it will not have resulted in adequate expression of channelrhodopsin in all of these cells.

We identified 8 tdTomato cells that were densely surrounded by GFP-labelled (Trpm8) afferents and found that 1 ms blue light pulses induced EPSCs in 5 of these cells (*Figure 3E and F*). We measured the latency of the first EPSC after blue light pulses (0.2 Hz) and found that the latency jitter was less than 1 ms, indicating that these neurons receive monosynaptic inputs (*Hachisuka et al., 2018*; *Figure 3F*; n=5, Average latency: 5.8±0.86 ms, latency jitter: 0.68±0.14 ms). Note that the relatively long latency is likely to reflect the time taken for action potential initiation in the Trpm8 boutons, as we have observed similar latencies when recording from cells receiving monosynaptic input from optogenetically activated neurons in this preparation (*Hachisuka et al., 2016*; *Hachisuka et al., 2020*). Two of the 5 neurons showed multiple peaks with a larger latency jitter, indicating that there may also have been polysynaptic inputs (*Figure 3—figure supplement 3*). It is likely that the remaining three cells received input from Trpm8 afferents that did not express sufficiently high levels of channelrhodopsin. Together, these findings demonstrate that Trpm8 afferents directly synapse onto lamina I ALS cells that are surrounded by these afferents.

## Lamina I ALS neurons that are surrounded by Trpm8 afferents are cold-selective

We next tested whether the lamina I projection neurons that are densely innervated by Trpm8 afferents correspond to the cold-selective neurons identified in physiological studies (*Craig and Kniffki, 1985*; *Craig et al., 2001*; *Craig and Dostrovsky, 2001*; *Craig and Hunsley, 1991*; *Dostrovsky and*

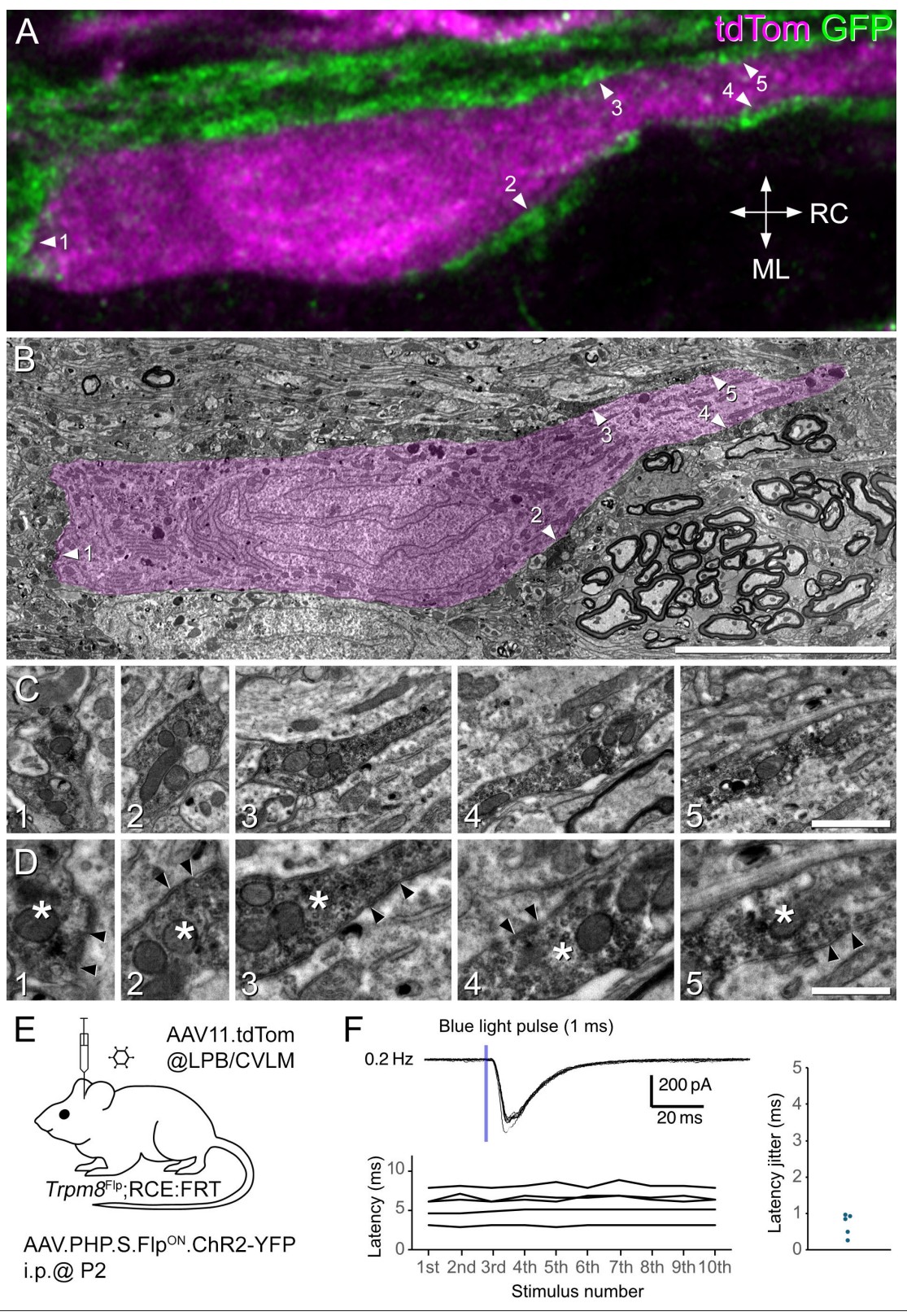

**Figure 3.** Evidence for monosynaptic input from Trpm8 afferents to lamina I anterolateral system (ALS) neurons that were surrounded by these afferents. (**A–D**) Combined confocal and electron microscopic examination of a lamina I projection neuron with numerous contacts from Trpm8-expressing afferents. (**A**) A confocal image (single optical section) from a horizontal section through lamina I, showing the cell body and part of a proximal dendrite of one of the tdTomato-expressing neurons that were analysed. The tissue was obtained from a *Phox2a*::Cre;Ai9;*Trpm8*^Flp^;RCE:FRT mouse. TdTomato

*Figure 3 continued on next page*

*Figure 3 continued*

expressed by the projection neuron is shown in magenta and GFP immunoreactivity in Trpm8 afferents in green. Several Trpm8 boutons contact the cell, and five of these are indicated with arrowheads. RC, rostro-caudal; ML, mediolateral. (**B**) A low magnification electron micrograph through the cell at approximately the same z-level as that shown in (**A**). Although the cell does not contain an electron dense label, it can be recognised by its shape, and because it is surrounded by GFP-expressing axonal profiles that were labelled with an immunoperoxidase method. The locations of those corresponding to the boutons shown in (**A**) are again marked with arrowheads. The cell has been pseudocoloured magenta to show its location. (**C,D**) Progressively higher magnification EM images showing the five GFP-labelled boutons marked with arrowheads in (**A**) and (**B**). In (**D**) the boutons are marked with asterisks and the locations of membrane darkening and vesicle clusters that presumably correspond to synapses on the cell body and dendrite of the projection neuron are indicated with arrowheads. (**E**) The experimental approach used for optogenetic testing of Trpm8 input to ALS neurons. Stimulation (1 ms, 0.2 Hz) evoked EPSCs in 5 of the ALS neurons that were densely coated by Trpm8 afferents. (**F**) shows an example of the response in one cell (upper trace), the latencies of EPSCs in response to 10 consecutive stimuli (n=5, lower graph), and on the right, the latency jitter for each of these cells (n=5). Scale bars: (**A,B**) = 10 μm, (**C**) = 1 μm (**D**) = 500 nm.

The online version of this article includes the following figure supplement(s) for figure 3:

**Figure supplement 1.** The relationship between tdTomato and GFP labelling for the cell illustrated in *Figure 3*.

**Figure supplement 2.** Confocal and electron microscope images of two Trpm8-innervated cells examined with the combined method in spinal cord sections from a *Phox2a*::Cre;Ai9;*Trpm8*^Flp^;RCE:FRT mouse.

**Figure supplement 3.** Optogenetic evidence for possible polysynaptic input from Trpm8 afferents to a lamina I anterolateral system (ALS) cell with dense Trpm8 input.

*Craig, 1996*; *Zhang et al., 2006*; *Allard, 2019*; *Andrew, 2009*; *Hachisuka et al., 2016*; *Hachisuka et al., 2020*; *Chisholm et al., 2021*). To do this, we carried out whole-cell patch-clamp recordings in preparations in which ALS neurons were retrogradely labelled with either mCherry or tdTomato, and Trpm8 axons expressed GFP. Retrograde labelling was achieved by injecting AAV9.mCherry into the CVLM or LPB of *Trpm8*^Flp^;RCE:FRT mice, resulting in mCherry labelling, or else by injecting AAV9.Cre_GFP into the CVLM of *Trpm8*^Flp^;RCE:FRT;Ai9 mice leading to tdTomato labelling (*Figure 4A*). We used the semi-intact ex vivo somatosensory preparation, in which the spinal cord, L2 and L3 roots and ganglia, saphenous and femoral cutaneous nerves, and hindlimb skin are dissected in continuity (*Figure 4B*; *Hachisuka et al., 2016*; *Hachisuka et al., 2020*).

Visualisation of the spinal cord with filter sets for GFP or mCherry/tdTomato revealed lamina I projection neurons (labelled with mCherry or tdTomato) as well as a plexus of GFP-labelled (Trpm8-expressing) primary afferents (*Figure 4*). We recorded from six mCherry/tdTomato-labeled projection neurons that had dendrites (and in some cases also cell bodies) that were clearly associated with numerous GFP-labelled afferents, as well as from five projection neurons for which there was no clear association with these afferents. We assessed the responses of these cells to a 4 g von Frey filament, hot saline (50 °C), and cold saline (15 °C) applied to the skin (*Hachisuka et al., 2016*). All six of the projection neurons that had numerous contacts from GFP-labelled axons responded exclusively to application of cold saline. In four cases recordings were made in current clamp and the cells fired action potentials (n=3) or generated EPSPs (subthreshold, n=1) following application of cold saline, but showed no response to mechanical or heat stimuli (*Figure 4C–E*). The time course of the response was consistent with that seen for cold-selective projection neurons that we previously reported (*Figure 4—figure supplement 1*; *Hachisuka et al., 2020*). The other two cells with numerous GFP contacts were recorded in voltage clamp, and these showed EPSCs in response to cold, but not mechanical or heat, stimuli. All six of these neurons were, therefore, classified as cold-selective. The five projection neurons that were not preferentially associated with GFP-labelled afferents were all recorded in current clamp mode, and all of these cells exhibited action potential firing in response to mechanical stimulation. Three of them also responded to heat, and one additionally to cold stimuli. The five neurons that were not associated with numerous GFP-labelled afferents were, therefore, classified as either mechano-selective or polymodal (*Figure 4F–H*).

We previously reported that cold-selective projection neurons are distinct from other lamina I projection neurons (*Hachisuka et al., 2020*), and a defining feature of these cells was their low frequency of spontaneous EPSCs (sEPSCs). We, therefore, quantified sEPSC frequency and amplitude in lamina I projection neurons with or without dense Trpm8 input. As expected, neurons with dense Trpm8 input had significantly lower sEPSC frequencies compared to those that lacked dense Trpm8 input (*Figure 4—figure supplement 1*). Interestingly, the amplitude of sEPSCs in the projection neurons with dense Trpm8 input was smaller than in those that lacked this input, which had not been

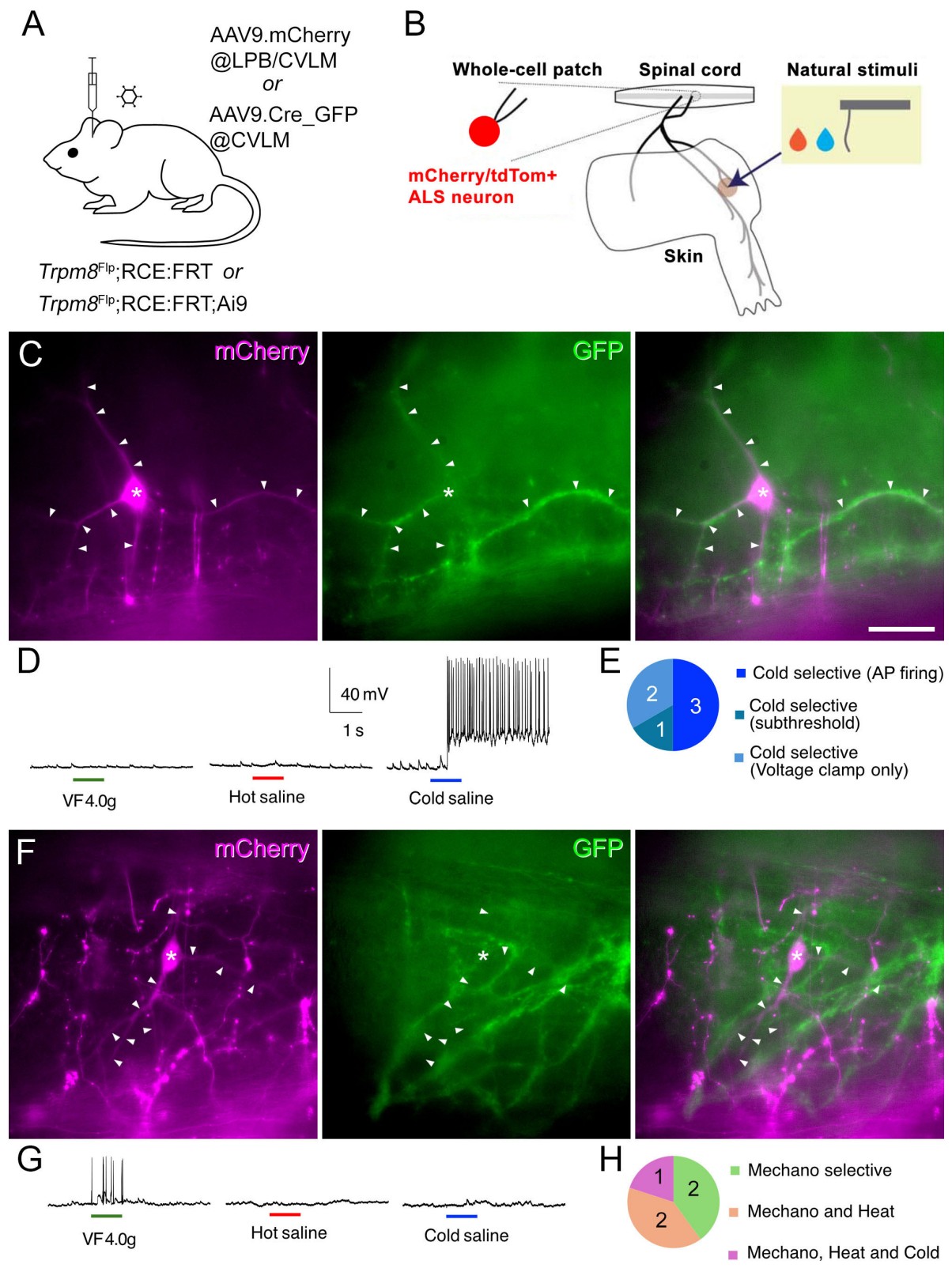

**Figure 4.** Electrophysiological characterisation of retrogradely labelled lamina I projection neurons recorded in *Trpm8*[Flp];RCE:FRT mice. (**A**) Cells were identified by expression of mCherry following injection of AAV9.mCherry into the caudal ventrolateral medulla (CVLM) or lateral parabrachial area (LPB) of *Trpm8*[Flp];RCE:FRT mice, or by expression of tdTomato following injection of AAV9. Cre_GFP into the CVLM of *Trpm8*[Flp];RCE:FRT;Ai9 mice. Note that the level of GFP expression resulting from injection of AAV9.Cre_GFP was extremely low, and would have been restricted to the nucleus of labelled

*Figure 4 continued on next page*

*Figure 4 continued*

neurons, due to the presence of a nuclear localisation signal. In both cases, Trpm8-expressing afferents were labelled with GFP. (**B**) The semi-intact somatosensory preparation retained skin attached to lumbar spinal cord through intact peripheral nerves and was used for recording responses to natural skin stimuli: Hot saline (50 °C), Cold saline (15 °C), or von Frey filaments (4 or 10 g) (**C**) Fluorescence microscopy image of a retrogradely labelled (mCherry-positive) lamina I ALS neuron (soma marked with asterisk) that is densely coated by Trpm8-expressing (GFP-positive) axons. Note that bundles of GFP axons are closely associated with the dendrites of the retrogradely labelled neuron, which are marked with arrowheads. (**D**) Whole-cell recording from this cell reveals that only cold stimulation evoked action potentials (APs). (**E**) Pie chart showing the response properties of the six lamina I ALS neurons that were densely coated with Trpm8 afferents to mechanical, cold, and heat stimulation of the skin. All of these cells responded only to cold stimulation: three cells showed APs, 1 cell showed EPSPs but no APs (subthreshold), while two cells were recorded in voltage clamp and showed EPSCs. (**F**) Fluorescence microscopy image of a lamina I ALS neuron that was not densely coated with Trpm8 axons. Again, the soma is marked with an asterisk and dendrites with arrowheads. Note that although the dendrites of this cell pass through bundles of GFP axons, these bundles are not aligned with the dendrites. (**G**) Whole-cell recording from the cell shown in (**F**). AP firing was evoked by mechanical stimulation (von Frey filament 4.0 g), but not by application of hot or cold saline to the skin. (**H**) Pie chart of response properties of 5 lamina I ALS neurons that lacked dense Trpm8 input to mechanical, cold, and heat stimulation to the skin. Two cells were mechano-selective, while three were polymodal (two cells: mechano, heat, one cell: mechano, heat, and cold). Scale bar (**C, F**) = 50 µm. Scale bars in (**D**) apply to (**G**).

The online version of this article includes the following figure supplement(s) for figure 4:

**Figure supplement 1.** Characteristics of spontaneous EPSCs in cold-selective and other lamina I anterolateral system (ALS) cells.

observed in our previous study (*Hachisuka et al., 2020*). Together, these results strongly suggest that lamina I projection neurons receiving dense Trpm8 afferent innervation constitute a distinct neuronal population specialised for cold sensation, and correspond to the cold-selective ALS neurons identified in previous studies (*Craig and Kniffki, 1985*; *Craig et al., 2001*; *Craig and Dostrovsky, 2001*; *Craig and Hunsley, 1991*; *Dostrovsky and Craig, 1996*; *Zhang et al., 2006*; *Allard, 2019*; *Andrew, 2009*; *Hachisuka et al., 2016*; *Hachisuka et al., 2020*; *Chisholm et al., 2021*).

Our anatomical studies indicate that the cells densely coated with Trpm8-expressing primary afferents receive many synapses from these afferents (*Figure 3*), and we had shown previously that these cells had numerous Homer1-immunoreactive puncta, with ~60% being associated with a Trpm8-expressing primary afferent bouton (*Bell et al., 2024*). The low frequency of sEPSCs is, therefore, not likely to result from a low density of synapses, but presumably reflects a very low release probability at these synapses in these experimental conditions. The lower amplitude of sEPSCs may be related to our EM observation that contacts between Trpm8 afferents and densely coated lamina I projection neurons typically have small synaptic specialisations (*Figure 3*, *Figure 3—figure supplement 2*).

## Targeting of cold-selective lamina I ALS neurons with the *Calb1*^Cre mouse

We had previously demonstrated that lamina I projection neurons that are densely innervated by Trpm8-expressing afferents are included among those of the Phox2a-lineage, and contain mRNA for *Hs3st1*, one of the top differentially expressed genes for ALS3. We had, therefore, assigned the Trpm8-innervated cells to the ALS3 cluster (*Bell et al., 2024*). *Calb1*, which encodes the calcium-binding protein calbindin, is also among the top 20 differentially expressed genes for ALS3 (*Figure 5—figure supplement 1A*, see also Figure 1D of *Bell et al., 2024*). However, calbindin is not restricted to the ALS3 cluster, as it is also expressed by some cells in the ALS2, ALS4, and ALS5 clusters (*Bell et al., 2024*; *Figure 5—figure supplement 1A*), and by many ALS neurons in the LSN (*Menétrey et al., 1992*), most of which are not derived from the Phox2a-lineage (*Roome et al., 2020*; *Alsulaiman et al., 2021*).

To assess the relationship between Calb1-expressing ALS neurons and those with dense Trpm8 innervation, we injected an AAV coding for Cre-dependent tdTomato (AAV11.Cre^ON.tdTomato) (*Han et al., 2023*) unilaterally into the LPB of 4 *Calb1*^Cre;*Trpm8*^Flp;RCE:FRT mice (*Figure 5—figure supplement 1B*). As expected from the known distribution of calbindin-expressing ALS cells in the rat (*Menétrey et al., 1992*), we observed tdTomato-positive cells in lamina I (mainly on the contralateral side), as well as bilaterally in the LSN and in deeper laminae (*Figure 5A*, *Figure 5—figure supplement 1C-F*). When we examined horizontal sections through lamina I, we found that many of the tdTomato-labelled cells were densely innervated by Trpm8 afferents (*Figure 5C–H*), and these were located throughout the mediolateral extent of the dorsal horn. The proportion of tdTomato cells with dense Trpm8 input varied between 38.7–59.6% across the L2-L4 segments (*Figure 5B*, RM ANOVA,

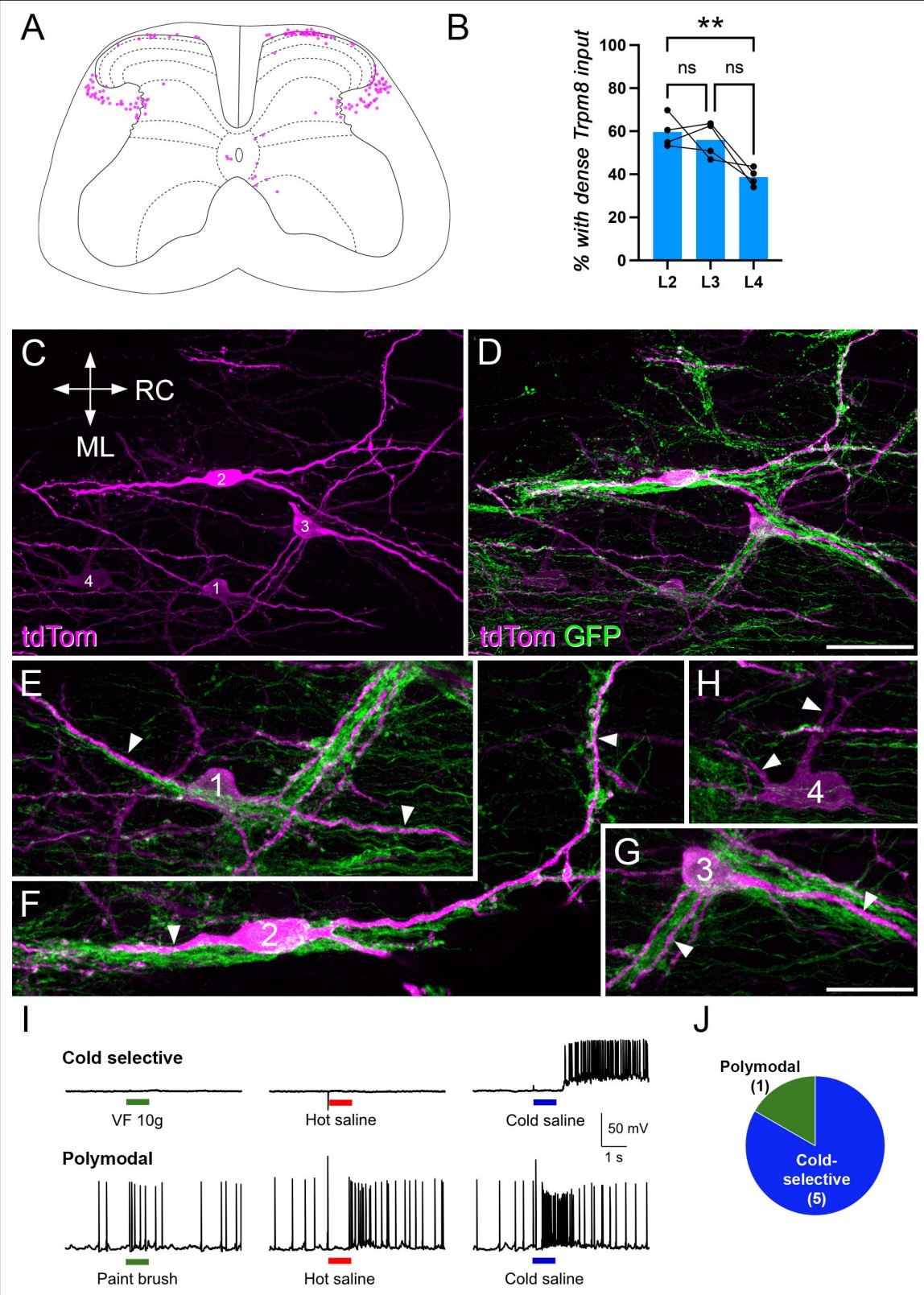

**Figure 5.** Characterisation of Calb1-expressing projection neurons. (**A**) The distribution of retrogradely labelled cells in 10 transverse sections from the L5 segment of one of the *Calb1*[Cre] mice that had received an injection of AAV11.Cre[ON].tdTomato into the lateral parabrachial area (LPB). The right side of the plot is the side contralateral to the brain injection. (**B**) Bar chart showing the proportion of retrogradely labelled cells in the L2, L3, and L4 segments of the *Calb1*[Cre] mice injected with AAV11.Cre[ON].tdTomato into the LPB that received dense Trpm8 input (n=4 mice). Repeated measures

*Figure 5 continued on next page*

*Figure 5 continued*

one-way ANOVA showed a significant difference (*p*=0.048), and post-hoc Holm-Šídák's multiple comparisons test revealed a difference only between L2 and L4 (*p*=0.0056). (**C**) Part of a horizontal section through the L2 segment of one of the mice, showing four lamina I neurons (numbered 1–4) that were retrogradely labelled with tdTomato (tdTom, magenta). RC, rostrocaudal; ML, mediolateral (applies to **C–H**). (**D**) The same field scanned to reveal both tdTomato and GFP (green). (**E–H**) detailed views of the four cells shown in (**C, D**). Three of these cells (1-3) have numerous contacts from GFP-labelled axons on their dendrites (marked with arrowheads) and cell bodies. The fourth cell (4) has very few such contacts. (**I**) Recordings from one of the cold-selective cells (upper traces) and the polymodal cell (lower traces) in experiments carried out in *Calb1*[Cre] mice that had received injections of AAV11. Cre[ON].tdTomato into the caudal ventrolateral medulla (CVLM). The cell in the upper traces shows action potential firing in response to application of cold saline to the skin, but no response to application of a 10 g von Frey (VF) hair or hot saline. The other cell responds to brushing of the skin with a paint brush, as well as to application of hot and cold saline. (**J**) Pie chart showing response characteristics of the six cells recorded in these experiments. Scale bars: (**C,D**) = 50 μm, (**E–H**) = 25 μm. \*\**p*<0.01.

The online version of this article includes the following figure supplement(s) for figure 5:

**Figure supplement 1.** Further characterisation of Calb1-expressing projection neurons.

**Figure supplement 2.** Electrophysiological characterisation of Calb1-expressing projection neurons.

*p*=0.048). Interestingly, the proportion was significantly higher in L2 (59.6%) than in L4 (38.7%), (RM ANOVA, Holm-Šídák's multiple comparisons test, *p*=0.0056, n=4), and this may reflect a difference in the proportion of lamina I cells with dense Trpm8 innervation between segments. When results across the 3 segments were pooled, the total numbers of retrogradely labelled cells ranged from 133 to 170 (mean 151), while the proportions with dense Trpm8 input ranged from 46.5–54.1% (mean 50.6%). To confirm expression of calbindin in lamina I projection neurons with dense Trpm8 input, we carried out immunofluorescence labelling on horizontal sections from 3 *Trpm8*[Flp];RCE:FRT mice that had received injections of AAV9.mCherry into the LPB. We identified a total of 31 mCherry-positive lamina I neurons that received numerous contacts from GFP-labelled axons (6–14 per mouse), and found that all but four of these (27/31, 87%) were calbindin-immunoreactive (*Figure 5—figure supplement 1G*).

We also used the semi-intact preparation to record from tdTomato-labelled lamina I neurons in spinal cords from 4 *Calb1*[Cre] mice that had received injections of AAV11.Cre[ON].tdTomato into the CVLM (*Figure 5—figure supplement 1B*, *Figure 5—figure supplement 2*). We recorded from six retrogradely labelled neurons and found that five were cold-selective, while one was polymodal, responding to brushing of the skin, as well as to application of both hot and cold stimuli (*Figure 5I and J*). These cold-selective neurons showed a similar time course in response to cold stimulation to those described above (*Figure 5—figure supplement 2C*). Taken together, these findings indicate that while Calb1 is not restricted to cold-selective cells, many of the Calb1-expressing projection neurons in lamina I belong to the cold-selective population.

## Brain projections of putative cold-selective lamina I neurons

Because Calb1 is expressed by cold-selective ALS cells in lamina I, we carried out anterograde tracing by injecting AAV coding for Cre-dependent expression of tdTomato (AAV1.Cre[ON].tdTom) into the lumbar spinal cords of *Calb1*[Cre] mice. Our aim was to test for projections to two potential brainstem targets that receive input from the ALS (*Wang et al., 2022*; *Wercberger and Basbaum, 2019*) and also contain many neurons that express the immediate early gene Fos in mice exposed to low ambient temperatures (*Cano et al., 2003*; *Geerling et al., 2016*; *Yang et al., 2023*; *Yoshida et al., 2005*): the caudal part of the PAG (cPAG), and the rostral part of the LPB, in particular a region that has been named PBrel (rostral to external lateral) (*Geerling et al., 2016*). Two cortical areas are activated by skin cooling, the primary somatosensory and posterior insular cortices (*Vestergaard et al., 2023*). We therefore looked for evidence of an input to the ventral posterolateral (VPL) and posterior triangular (PoT) thalamic nuclei, which are thought to provide the thermosensory inputs to these cortical areas in rodents (*Leva and Whitmire, 2023*). Initially, we examined labelling resulting from injections of AAV1. Cre[ON].tdTomato that were targeted on the central part of the dorsal horn, either in the L3 segment, or in the L3, L4, and L5 segments, using a similar approach to that which we had taken with other genotypes (*Figure 6—figure supplement 1A and B*; *Table 1*). Labelling patterns were consistent among the mice used in this part of the study, but labelling was denser in all regions in those that had received injections into L3, L4, and L5. This approach revealed that axons ascended in the dorsolateral white matter of the spinal cord and gave branches to several regions on the contralateral side of the brainstem, including LPB and the cPAG (*Figure 6—figure supplement 1C and D*). Within the LPB,

**Table 1.** Experimental details for *Calb1*[Cre] mice used in anterograde tracing experiments.

| Animal | Sex | Injected segments | Injection location | Injection volume (nl) | Viral load (GC) | Survival (days) |
|---|---|---|---|---|---|---|
| 1 | M | L3, L4, L5 | central | 300 | $9.48\times10^7$ | 26 |
| 2 | M | L3, L4, L5 | central | 300 | $9.48\times10^7$ | 55 |
| 3 | F | L3, L4, L5 | central | 300 | $9.48\times10^7$ | 57 |
| 4 | M | L3 | central | 300 | $9.48\times10^7$ | 42 |
| 5 | F | L3 | central | 300 | $9.48\times0^7$ | 40 |
| 6 | F | L3 | central | 300 | $9.48\times10^7$ | 40 |
| 7 | M | L3, L4, L5 | medial | 150 | $4.74\times10^7$ | 55 |
| 8 | M | L3, L4, L5 | medial | 150 | $4.74\times10^7$ | 54 |
| 9 | F | L3, L4, L5 | medial | 150 | $4.74\times10^7$ | 41 |
| 10 | F | L3, L4, L5 | medial | 150 | $4.74\times10^7$ | 55 |
| 11 | F | L3 | medial | 150 | $4.74\times10^7$ | 41 |
| 12 | F | L3 | medial | 150 | $4.74\times10^7$ | 41 |
| 13 | F | L3 | medial | 150 | $4.74\times10^7$ | 53 |
| 14 | M | C8 | central | 500 | $1.58\times10^8$ | 35 |
| 15 | M | C8 | central | 500 | $1.58\times10^8$ | 44 |
| 16 | M | C8 | central | 500 | $1.58\times10^8$ | 43 |

Mice received injections of AAV1.Cre[ON].tdTomato into the L3 segment only, the L3, L4, and L5 segments, or the C8 segments on the right side. Injections in lumbar cord were either targeted centrally (400 µm lateral to the midline) or medially (250 µm lateral to the midline) within the dorsal horn. Those in the cervical cord were located centrally (430–450 µm lateral to the midline). All injections were made at 300 µm below the surface of the spinal cord. Injection volume and viral load refer to each injection in animals 1–3 and 7–9.

axonal labelling was very dense in the rostral-most part, which corresponds to PBrel (*Geerling et al., 2016*), and moderately dense in the centrolateral (PBcl) nucleus. In addition, there was some labelling in other nuclei, including external lateral (PBel), dorsolateral (PBdl), and internal lateral (PBil). There was dense labelling in the cPAG and in addition, labelling was seen within the superior colliculus (*Figure 6—figure supplement 1C*). In the thalamus, there was sparse labelling in the VPL, posterior (Po), and PoT nuclei contralateral to the injection site, and many labelled profiles were present in the medial thalamus on both sides (*Figure 6—figure supplement 1E*). These findings were, therefore, consistent with the expected projections of cold-selective ALS cells to PBrel, cPAG, and to the PoT and VPL nuclei of the thalamus.

However, this strategy will also have labelled Calb1-expressing projection neurons in other regions, including the LSN and the lateral reticulated part of lamina V (*Figure 6—figure supplement 1B*). Consistent with this interpretation, we observed labelling in PBil and the medial thalamus, both of which are known to receive input from ALS cells in the deep dorsal horn (*Bernard et al., 1995*; *Feil and Herbert, 1995*; *Gauriau and Bernard, 2004*), including cells belonging to the ALS4 population in lateral lamina V (*Ma et al., 2025b*). We, therefore, altered the injection technique by reducing the volume and injecting at a more medial location (250 µm rather than 400 µm lateral to the midline). As with the previous experiments, these injections were either made into the L3 segment, or into each of L3, L4, and L5 (*Table 1*). Again, the pattern of labelling was consistent among these mice, but labelling was denser in those that had received injections into L3, L4, and L5. This strategy resulted in injection sites that were restricted to the medial part of the dorsal horn and excluded the LSN (*Figure 6A–C*). Although very weak tdTomato labelling was seen in the LSN in these experiments, this is likely to have arisen from transport by axons of local calbindin-expressing excitatory interneurons, since many excitatory interneurons in laminae I-II have axons that enter the LSN (*Gutierrez-Mecinas et al., 2018*; *Gutierrez-Mecinas et al., 2019*; *Polgár et al., 2023*). Importantly, no tdTomato-labelled cell bodies were seen in the LSN or in the lateral reticulated part of lamina V in these cases. Since these injections

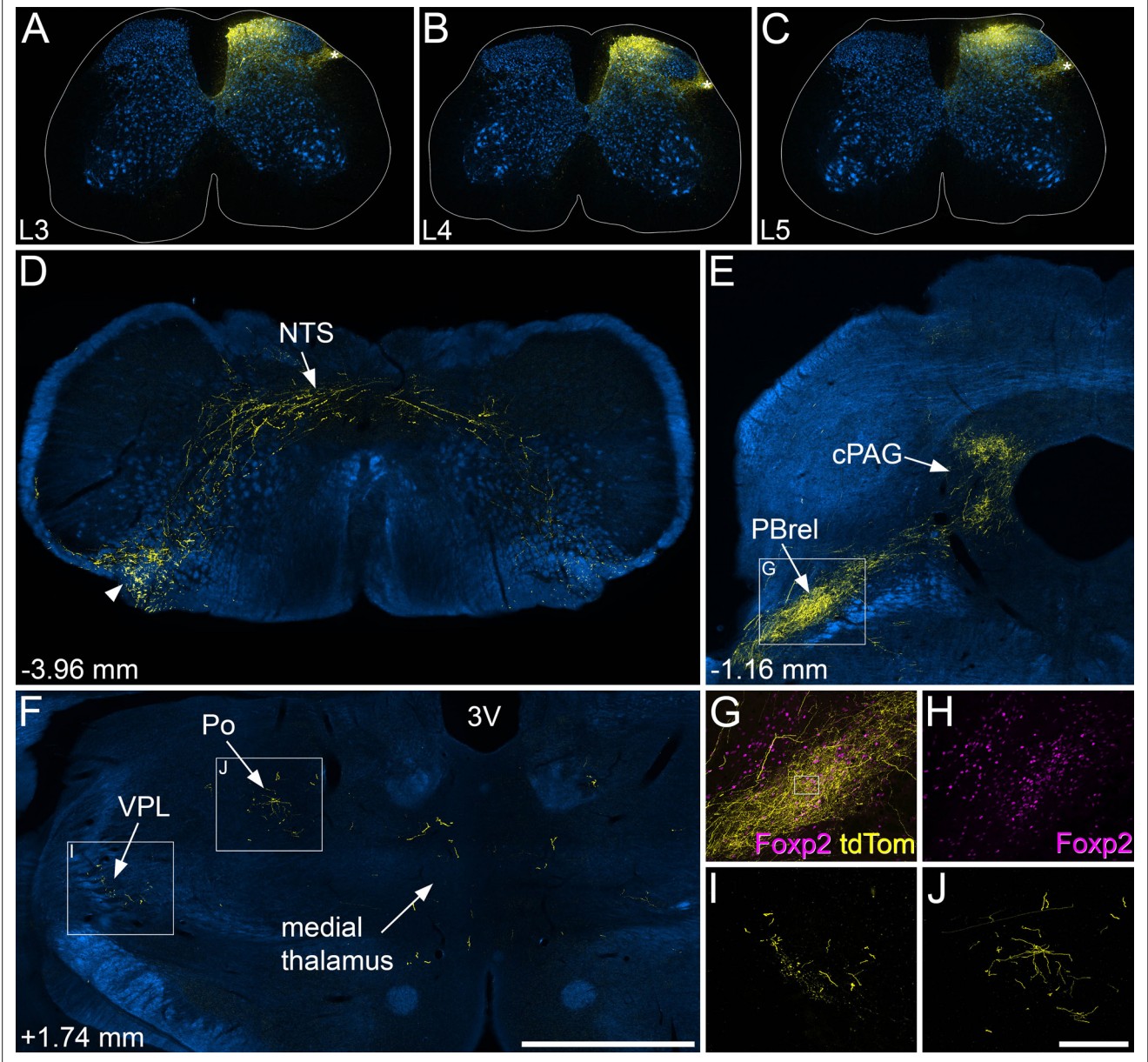

**Figure 6.** Anterograde tracing following injections of AAV1.Cre^ON.tdTomato into the medial dorsal horn of *Calb1*^Cre mice. (**A–C**) Injection sites in the medial parts of the L3, L4, and L5 segments of one of the mice used in this part of the study. Immunostaining for tdTomato is shown in yellow and for NeuN in blue. Labelling of cell bodies is largely restricted to the medial part of the dorsal horn. There is some labelling in the lateral spinal nucleus (LSN), marked by asterisks, but this is likely to result from transport in axons of local interneurons. (**D–F**) Sections through selected regions of the brain immunostained to reveal tdTomato (yellow), with the corresponding dark-field images shown in blue. (**D**) At the level of the caudal medulla, the main ascending fibre bundle (arrowhead) lies in the ventrolateral region on the side contralateral to the spinal injections. Collateral axons innervate the nucleus of the solitary tract (NTS). (**E**) There is a dense projection of axons to the rostralmost part of the lateral parabrachial area (PBrel) and the caudal part of the PAG (cPAG) on the side contralateral to the injections. (**F**) Within the diencephalon there is sparse labelling in the contralateral ventral posterolateral nucleus (VPL) and posterior nucleus (Po) as well as within the medial thalamus. (**G, H**) The region indicated in the box in (**E**) is shown at higher magnification with immunostaining for Foxp2 shown in magenta. The overlap between arborisation of tdTomato and Foxp2-positive nuclei indicates that this region is indeed PBrel. The box in (**G**) corresponds to the region shown at higher magnification in *Figure 6—figure supplement 2F*. (**I, J**) show higher magnification views of tdTomato-positive axons in the VPL and Po, respectively, corresponding to the boxes in (**F**). Numbers in (**D–F**) show approximate rostrocaudal locations in relation to the interaural line. Images in (**A–E, G–H**) are from animal #8, and those in (**F, I, J**) are from animal #7 in *Table 1*. Scale bars: (**A–F**) = 1 mm, (**G–J**) = 200 µm.

The online version of this article includes the following figure supplement(s) for figure 6:

*Figure 6 continued on next page*

*Figure 6 continued*

**Figure supplement 1.** Anterograde labelling following injections of AAV1.Cre[ON].tdTomato into the central part of the spinal dorsal horn of *Calb1*[Cre] mice.

**Figure supplement 2.** Anterograde labelling following injections of AAV.Cre[ON].tdTomato into the medial dorsal horn of *Calb1*[Cre] mice.

**Figure supplement 3.** Anterograde labelling following injections of AAV.Cre[ON].tdTomato into the cervical dorsal horn of *Calb1*[Cre] mice.

**Figure supplement 4.** Retrograde labelling of Calb1-expressing anterolateral system (ALS) neurons from periaqueductal grey matter (PAG).

**Figure supplement 5.** Identification of spinothalamic lamina I neurons with dense Trpm8 input.

were likely to have included a much higher proportion of lamina I neurons with dense Trpm8 innervation, we analysed the anterograde labelling in more detail.

Axons ascended in the dorsal quadrant of the spinal cord, mainly on the contralateral side (***Figure 6—figure supplement 2A***). Although a few collaterals arose at all segmental levels examined, these were far less numerous than those originating from the somatostatin-expressing ALS cells that occupy the lateral part of lamina V (***Ma et al., 2025b***). Ascending axons lay near the ventrolateral surface of the medulla, and sent collateral branches to the NTS (***Figure 6D***), but unlike Gpr83- and Tacr1-expressing projection neurons (***Ma et al., 2025b***; ***Choi et al., 2020***), these Calb1-positive cells did not innervate the inferior olive. Labelling in the LPB was much more restricted following medial spinal injections, compared to the central injections (***Figure 6E, F***, ***Figure 6—figure supplement 1C,D***, ***Figure 6—figure supplement 2B***). In particular, labelling in PBcl and PBil was greatly reduced, while the dense input to the PBrel was retained. The input to PBrel was confirmed by immunostaining for the transcription factor Foxp2, which is expressed in PBrel neurons (***Figure 6G and H***; ***Geerling et al., 2016***). There was also strong labelling in the lateral part of the cPAG, (***Figure 6E***), but only very sparse labelling in the rostral part of the PAG (***Figure 6—figure supplement 2C and E***). Input to both LPB and PAG was bilateral, but denser on the side contralateral to the spinal injections. Labelling in the superior colliculus was almost completely lost following medial spinal injections (***Figure 6E***). Within the thalamus, the sparse labelling in VPL and Po was still present, but that in the medial thalamus was far weaker (***Figure 6F, I and J***). In addition, medial spinal injections resulted in labelling in the PoT nucleus of the thalamus (***Figure 6—figure supplement 2C and D***), which is a known target for lamina I ALS neurons (***Gauriau and Bernard, 2004***; ***Al-Khater et al., 2008***) and is thought to convey non-noxious thermal information to the posterior insular cortex (***Leva and Whitmire, 2023***). Very few, if any, tdTomato-labelled axons were seen in any other parts of the diencephalon (***Franklin and Paxinos, 2007***). High magnification scans showed that in the regions targeted by tdTomato labelled axons (NTS, LPB, PAG, thalamus), these axons possessed numerous varicosities, which presumably correspond to axonal boutons (***Figure 6—figure supplement 2F***). Immunostaining for the postsynaptic density protein Homer1 revealed that varicosities were often apposed to Homer1 puncta (***Figure 6—figure supplement 2G and H***), confirming that they constituted the presynaptic elements at glutamatergic synapses.

We have previously demonstrated that although spinothalamic lamina I neurons are relatively common in the rodent cervical enlargement, they are far less numerous at lumbar levels, accounting for <5% of lamina I ALS neurons (***Alsulaiman et al., 2021***; ***Al-Khater and Todd, 2009***). We, therefore, injected AAV1.Cre[ON].tdTom into the cervical enlargement of 3 *Calb1*[Cre] mice (***Table 1***, ***Figure 6—figure supplement 3A and B***). This resulted in far denser axonal labelling within the VPL and PoT nuclei on the side contralateral to the spinal injection (***Figure 6—figure supplement 3C–E***).

The differences between the projection patterns of medial and central injections into the lumbar enlargement provide additional information about potential targets of Calb1 cells that were not located in lamina I. As noted above, the input to PBil and medial thalamus seen following the central injections is likely to result at least in part from capture of cells belonging to the ALS4 population in lateral lamina V (***Bell et al., 2024***), since we have shown that these cells (targeted in a *Sst*[Cre] mouse line) project to these two sites (***Ma et al., 2025b***). However, we found only very sparse input from the Sst-expressing (ALS4) cells to the superior colliculus (***Ma et al., 2025b***), raising the possibility that input to the colliculus originates mainly from cells in the LSN, many of which express Calb1. Consistent with this suggestion, ***Choi et al., 2020*** observed a projection to the superior colliculus from Tacr1-expressing ALS cells, and many ALS neurons in the LSN express the neurokinin 1 receptor (NK1r), which is encoded by Tacr1 (***Ma et al., 2025a***; ***Ding et al., 1995***; ***Marshall et al., 1996***).

To confirm the input to PAG from cold-selective lamina I cells, we targeted injections of AAV11. Cre[ON].tdTomato to the caudal PAG in 3 *Calb1*[Cre];*Trpm8*[Flp];RCE:FRT mice. In these experiments, the total numbers of retrogradely labelled (tdTomato-positive) lamina I cells in the L2-L4 segments were lower than was the case for the LPB injection experiments (26, 59, and 99 in the three cases), but the proportion of these cells that had dense Trpm8 input was significantly higher: 84.6%, 81.4%, and 77.8% ($p<0.0001$, unpaired t-test with Welch's correction; *Figure 6—figure supplement 4*). This indicates that, at least for calbindin-expressing lamina I ALS neurons, cold-selective cells are highly enriched among those that project to the caudal part of the PAG.

Finally, we looked for further evidence of a projection from cold-selective lamina I cells to the lateral part of the thalamus by making small injections of cholera toxin B subunit (CTB) in 3 *Trpm8*[Flp];RCE:FRT mice. CTB was used as a retrograde tracer for these experiments, because it can result in relatively small and well-defined injection sites. In all three cases, the injection was centred on the ventral posterior nucleus of thalamus, and included both lateral (VPL) and medial (VPM) parts (*Figure 6—figure supplement 5A–C*). There was some spread into Po and the ventrolateral (VL) nucleus, but none into the medial thalamus and none of the injections extended as far caudally as the PoT. As expected, from the low number of lamina I spinothalamic neurons at lumbar levels (*Alsulaiman et al., 2021*; *Al-Khater and Todd, 2009*), few (if any) retrogradely labelled cells were seen in the lumbar spinal cord in these experiments (five cells in the L4 segment of one mouse, but no cells in this segment in the other two), and for this reason, we focused our search on the cervical enlargement. In horizontal sections through the C7 segment, we identified a large number of retrogradely labelled lamina I cells in each case. Although the dendritic filling was less complete than that seen with retrograde viral tracing, we were able to identify many CTB-positive cells that received numerous contacts on cell bodies and dendrites from Trpm8-expressing primary afferents (*Figure 6—figure supplement 5D–I*). In the one case in which retrogradely labelled cells were detected in the L4 segment, 3 of the 5 cells were also densely coated with GFP-labelled afferents (*Figure 6—figure supplement 5J–L*). These findings demonstrate that cold-selective lamina I ALS neurons project to the lateral thalamus.

## Discussion

The main findings of this study are that: (1) lamina I projection neurons that are densely innervated by Trpm8-expressing primary afferents correspond to the cold-selective cells identified in physiological studies and receive monosynaptic input from Trpm8 afferents; (2) these are captured when AAVs encoding Cre-dependent constructs are injected into the brains of *Calb1*[Cre] mice; and (3) axons of Calb1-positive neurons in the medial part of the dorsal horn project to brain regions known to be involved in processing information related to cold input, PBrel, caudal PAG, as well as the PoT and VPL nuclei of the thalamus. This suggests that these sites are directly innervated by cold-selective ALS neurons in lamina I.

### Lamina I ALS neurons with dense Trpm8 input are cold-selective

We had previously observed that Trpm8-expressing primary afferents are closely associated with the cell bodies and dendrites of a subset of ALS cells in lamina I *Bell et al., 2024*. Based on their association with Homer1 puncta, we were also able to estimate that GFP-labelled (Trpm8-expressing) afferents accounted for ~60% of the excitatory synapses on these ALS cells. Here, we directly demonstrate the presence of synapses with both electron microscopy and optogenetics, and confirm our prediction (*Bell et al., 2024*) that these densely Trpm8-innervated ALS cells correspond to the cold-selective neurons identified in previous physiological studies.

Cold-selective lamina I projection neurons have been reported in several species, including monkey (*Dostrovsky and Craig, 1996*), cat (*Craig and Kniffki, 1985*; *Craig et al., 2001*; *Craig and Dostrovsky, 2001*; *Craig and Hunsley, 1991*), rat (*Zhang et al., 2006*; *Andrew, 2009*), and mouse (*Allard, 2019*; *Hachisuka et al., 2016*; *Hachisuka et al., 2020*; *Chisholm et al., 2021*), and a recent in vivo calcium-imaging study reported that they accounted for ~15% of lamina I spinoparabrachial neurons in the mouse (*Chisholm et al., 2021*). Although cold-selective lamina I projection neurons respond predominantly to skin cooling, some show a weak 'paradoxical' response when noxious heat is applied to the skin (*Craig et al., 2001*; *Craig and Hunsley, 1991*; *Dostrovsky and Craig, 1996*; *Allard, 2019*). We did not detect any response from cold-selective cells when hot saline was applied

to the skin in the experiments reported here, or in our previous study of cold-selective neurons (*Hachisuka et al., 2020*), and this may be because although the temperature of the hot saline was 50 °C, the small volume applied to the skin meant that skin temperature did not reach a sufficiently high level to result in activation. It is known that some Trpm8-expressing primary afferents have low levels of Trpv1 (*Takashima et al., 2007*; *Qi et al., 2024*; *Zeisel et al., 2018*), and some Trpm8 afferents have been shown to respond weakly to noxious heat in vivo (*Qi et al., 2024*). It has been proposed that there are two different populations of Trpm8-expressing afferents: one consisting of cells that lack Trpv1 and respond to innocuous cooling, and the other of cells that express Trpv1, and may function as 'cold nociceptors' (*McKemy, 2013*). Since we found Trpv1-immunoreactivity in around a quarter of GFP-labelled dorsal root ganglion cells, it is likely that both populations are included amongst the GFP-expressing cells in the *Trpm8*$^{Flp}$;RCE:FRT mice. However, we were unable to detect Trpv1-immunoreactivity in the central terminals of any GFP-positive axons (presumably reflecting the low level of Trpv1 expressed by these cells), and we therefore do not know whether the Trpm8-positive axons synapsing on cold-selective lamina I ALS neurons include those that express Trpv1. However, if this is the case, it is likely to explain the weak response to noxious heat reported for some cold-selective projection neurons (*Craig et al., 2001*; *Craig and Hunsley, 1991*; *Dostrovsky and Craig, 1996*; *Allard, 2019*).

## Direct and indirect input from Trpm8 afferents to cold-selective ALS neurons

Our findings clearly demonstrate a major direct (monosynaptic) input from Trpm8-expressing primary afferents to cold-selective lamina I projection neurons. We had previously estimated that a mean of 62% of the Homer1 puncta (which correspond to excitatory synapses) on cell bodies and dendrites of these cells were associated with GFP-positive boutons in *Trpm8*$^{Flp}$:RCE:FRT mice (*Bell et al., 2024*). However, we show here that this line only labels ~83% of Trpm8-expressing primary afferent neurons with GFP. If this proportion applies equally to Trpm8 afferents that synapse on cold-selective ALS cells, then Trpm8 afferents would give rise to ~75% of the excitatory synapses on these cells.

*Lee et al., 2025* recently proposed that there is a dedicated population of superficial dorsal horn excitatory interneurons that express the thyrotropin-releasing hormone receptor (Trhr) and form a disynaptic link between Trpm8 afferents and cold-selective lamina I projection neurons. Interestingly, in their initial experiments, they ablated or silenced calbindin-expressing dorsal horn neurons and found that this reduced behavioural responses to skin cooling. Although they interpreted this as resulting from loss of function of calbindin-expressing excitatory interneurons, our findings indicate that they are also likely to have ablated or silenced calbindin-expressing cold-selective lamina I ALS neurons. This could, therefore, have made a major contribution to the reduced sensitivity to skin cooling. Having obtained evidence that cold-responsive Calb1-positive neurons in the dorsal horn often co-expressed Trhr, they generated a *Trhr*$^{Cre}$ mouse line and used spinal injection of AAVs coding for Cre-dependent constructs, or a genetic intersectional approach, to target Trhr-expressing spinal neurons (*Lee et al., 2025*). They ablated these cells and found that this selectively reduced responses to skin cooling. They also reported that when AAV.Cre$^{ON}$.YFP was injected intraspinally in *Trhr*$^{Cre}$ mice, there was no axonal labelling in the brain, suggesting that Trhr-expressing dorsal horn neurons were exclusively interneurons. However, this observation is not consistent with our transcriptomic dataset (*Bell et al., 2024*), which reveals Trhr expression in some Phox2a-derived ALS neurons, including some of those in the ALS3 cluster (*Figure 7—figure supplement 1*). It is, therefore, not clear why a brain projection from Trhr cells was not detected by Lee et al, although it is possible that the survival time allowed for anterograde transport of YFP (reported as 2–3 weeks) was not sufficient. *Lee et al., 2025* also reported that the cold-selective ALS neurons were marked by expression of the calcitonin receptor-like receptor (encoded by the *Calcrl* gene). However, this is unlikely to be a specific marker, as our data show expression of Calcrl among several populations of Phox2a-derived ALS neurons (*Bell et al., 2024*; *Figure 7—figure supplement 1*), and it has been reported that Calcrl is also expressed by spinoparabrachial neurons that are required for mechanical itch (*Ren et al., 2023*).

While our findings do not exclude the possibility that there is a specific population of interneurons that link Trpm8-expressing primary afferents to cold-selective ALS neurons in lamina I, they clearly demonstrate a substantial direct (monosynaptic) input from Trpm8 afferents to cold-selective lamina I

projection neurons, and indicate the need for caution in relying on the specificity of individual genetic markers to define neurons and circuits in the dorsal horn.

## Brain targets of cold-selective ALS cells

At present, there is no known molecular marker that could be used to target cold-selective lamina I ALS cells exclusively. We chose calbindin because *Calb1* is among the top differentially expressed genes in the ALS3 cluster (*Bell et al., 2024*), and there is a well-established knock-in Cre line available (*Madisen et al., 2010*). Importantly, our transcriptomic analysis (*Bell et al., 2024*) was restricted to projection neurons derived from the Phox2a lineage (*Roome et al., 2020*), which excludes most ALS cells in the LSN and ~40% of those in lamina I (*Roome et al., 2020*; *Alsulaiman et al., 2021*). *Menétrey et al., 1992* had demonstrated that calbindin-immunoreactive spinoparabrachial neurons in rat were located in lamina I, the LSN, and lateral lamina V, with scattered cells around the central canal, and this closely matches the distribution of cells labelled when AAV11.Cre$^{ON}$.tdTomato was injected into the LPB of *Calb1*$^{Cre}$ mice. For anterograde tracing, we were able to exclude LSN and lateral lamina V ALS cells by injecting smaller volumes into the medial part of the dorsal horn, and this was confirmed by the lack of axonal labelling in regions known to receive input from these cells (e.g. PBil and medial thalamus) (*Ma et al., 2025b*; *Bernard et al., 1995*; *Gauriau and Bernard, 2004*). However, although our medial injections preferentially captured cold-selective lamina I ALS neurons, they will also have labelled cells that are not cold-selective, including some lamina I cells in the ALS2 cluster (*Bell et al., 2024*) and potentially some of those that are not derived from the Phox2a lineage (*Alsulaiman et al., 2021*). We, therefore, focused our analysis on brain regions known to be involved in cold sensing, and tested whether these received input from the Calb1-positive ALS cells.

Among the ALS projection targets that contain cells responding to cold, we were able to show a dense input to the rostral part of the LPB and to the cPAG. Two studies have mapped neurons in LPB that express Fos in mice exposed to low ambient temperatures (*Geerling et al., 2016*; *Yang et al., 2023*), and both found that Fos-positive cells were concentrated in the rostral part of the LPB, corresponding to a level ~4.9 mm caudal to bregma (or 1.1 mm caudal to the interaural line) (*Franklin and Paxinos, 2007*). *Yang et al., 2023* identified the region containing Fos-positive cells as part of the external lateral (PBel) nucleus, while *Geerling et al., 2016* defined it as the PBrel, and showed that this area could be recognised by the expression of Foxp2. *Yang et al., 2023* reported that cold-activated cells in this region projected to the preoptic area (POA), which is known to form part of a cold-defence circuit (*Morrison, 2018*), as well as to the dorsomedial hypothalamus (DMH). They also showed that parabrachial neurons targeting DMH promoted activation of brown adipose tissue, suggesting a role for both of these hypothalamic areas in maintenance of body temperature in cold conditions. The cold-selective lamina I ALS cells are therefore likely to provide an important source of input to POA and DMH via their projection to PBrel, thus forming part of the afferent limb of a cold defence pathway (*Figure 7*).

Neurons in the cPAG show increased Fos expression in rats exposed to low ambient temperatures (*Cano et al., 2003*), and this region receives an input from cold-activated cells in the hypothalamus (*Yoshida et al., 2005*). Our results demonstrate that Calb1-positive neurons in the dorsal horn also provide a dense input to the cPAG. Although we cannot be certain that this originates from the cold-selective ALS neurons, there are two lines of evidence to support this suggestion. First, *Li et al., 2023* demonstrated innervation of mouse lamina I spino-PAG neurons by Trpm8-expressing primary afferents, and second, we found that ~80% of Calb1 neurons retrogradely labelled from cPAG received dense Trpm8 input. Consistent with this interpretation, we had previously reported that rat lamina I neurons retrogradely labelled from the PAG seldom showed strong expression of the NK1r, unlike those labelled from the LPB or CVLM (*Spike et al., 2003*), and cold-selective lamina I neurons have been shown to respond weakly, if at all, to substance P, the main ligand for this receptor (*Hachisuka et al., 2020*). These findings suggest that the cold-selective lamina I neurons provide a direct input to cPAG, which presumably contributes to the responsiveness of neurons in this region to cooling (*Cano et al., 2003*). Activation of neurons in the cPAG results in brown adipose tissue thermogenesis (*Chen et al., 2002*), suggesting a role in maintenance of body temperature in cold conditions (*Figure 7*).

Our CTB injections into the lateral thalamus were centred on the ventral posterior nucleus, with a varying degree of spread into Po, which was also found to receive input from Calb1-positive ALS neurons. It is not possible to determine precisely the region(s) from which retrograde transport would

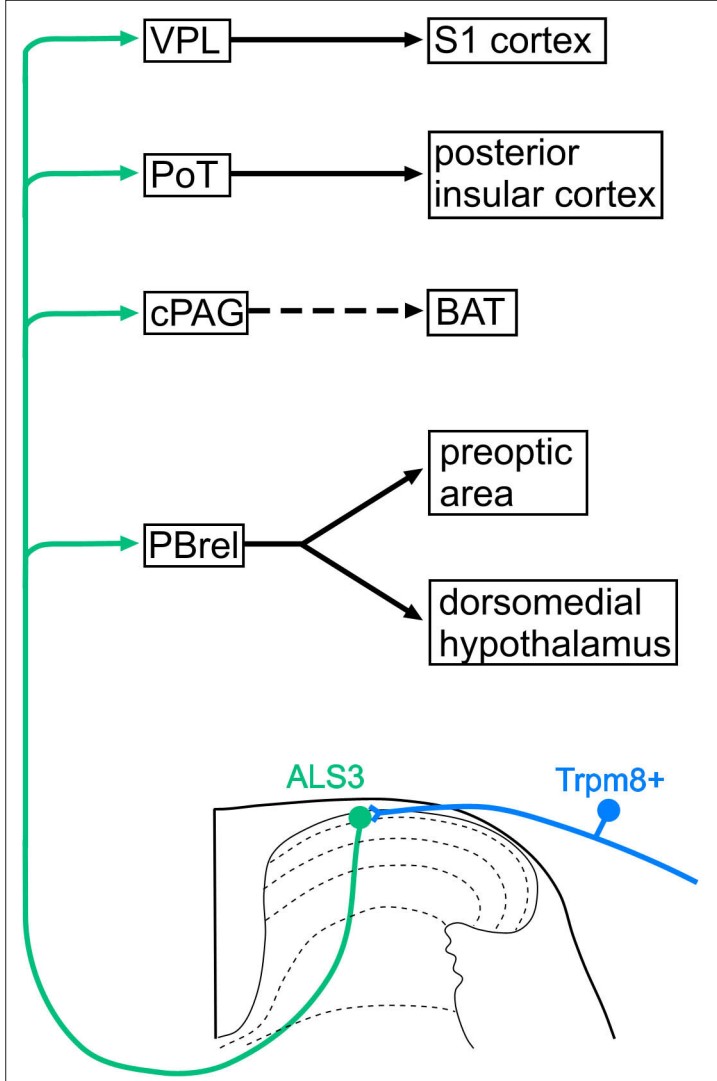

**Figure 7.** Proposed circuits involving cold-selective lamina I anterolateral system (ALS) neurons. Our results suggest that cold-selective lamina I projection neurons, which belong to the ALS3 cluster (***Bell et al., 2024***) project, among other sites, to PBrel, the caudal part of the PAG (caudal part of the periaqueductal grey matter, cPAG), and the ventral posterolateral nucleus of the thalamus (VPL). Cold-responsive cells in PBrel innervate the preoptic area and dorsomedial hypothalamus, which are integral components of circuits that underlie cold defence. The cPAG can indirectly activate brown adipose tissue (BAT) metabolism, also contributing to cold defence. The projection to posterior triangular (PoT) and VPL, through their connections to the posterior insular and primary somatosensory (S1) cortices, presumably underlies conscious perception of cold stimuli applied to the skin. Although we show a single ALS3 neuron projecting to different brain regions, it is possible that branches to different brain nuclei originate from specific subsets of ALS3 neurons. Note that in addition to the direct (monosynaptic) input from Trpm8-expressing primary afferents to ALS3 neurons, there may also be indirect (e.g. polysynaptic) inputs arriving via excitatory interneurons.

The online version of this article includes the following figure supplement(s) for figure 7:

**Figure supplement 1.** UMAP plot showing the distribution of cells in the ALS1-5 clusters, together with plots for expression of thyrotropin releasing hormone receptor (Trhr) and Calcrl.

---

occur, but since we identified lamina I cells in the C7 segment (and in one case also in the L4 segment) with relatively strong CTB labelling that were associated with GFP-labelled axons, these are likely to have projected to the core of the injection site, i.e., the ventral posterior nucleus. As noted above, spinothalamic lamina I neurons are infrequent in the rodent lumbar enlargement (***Alsulaiman et al., 2021***; ***Al-Khater and Todd, 2009***), and this presumably accounts for the very sparse labelling in VPL

that we saw following injections of AAV1.Cre^ON.tdTomato into the lumbar spinal cord, compared to that seen after cervical injections. However, our findings from the retrograde tracing experiments suggest that cold-selective cells project to VPL, and this is consistent with the demonstration of input from cervical lamina I neurons to VPL in the rat (*Gauriau and Bernard, 2004*). In addition, we identified an input to the PoT nucleus of thalamus, which projects to the posterior insular cortex (*Leva and Whitmire, 2023*). Together, the input from cold-selective lamina I ALS cells to PoT and VPL is likely to contribute to the perception of cold stimuli applied to the skin through the projections of these thalamic nuclei to the posterior insular and somatosensory cortices (*Leva and Whitmire, 2023*; *Figure 7*).

## Materials and methods

### Experimental model and subject details

All experiments were approved by the Animal Welfare and Ethical Review Body of the University of Glasgow, and were carried out in accordance with the UK Animals (Scientific Procedures) Act 1986 and ARRIVE guidelines. The following transgenic mouse lines were used in this study: the BAC transgenic *Phox2a*::Cre line, which expresses Cre recombinase under control of the *Phox2a* promoter (*Roome et al., 2020*); the Ai9 Cre reporter line, in which a loxP-flanked STOP cassette prevents CAG promoter-driven transcription of tdTomato (*Madisen et al., 2010*); the *Trpm8*^Flp line, in which 2A-linked FlpO recombinase is fused with the last codon (and replaces the stop codon) of the *Trpm8* gene (*Bell et al., 2024*); the RCE:FRT line, in which a FRT-flanked STOP cassette prevents CAG promoter-driven transcription of EGFP (*Sousa et al., 2009*); and the *Calb1*^Cre line, which has an IRES2 sequence and a Cre recombinase gene inserted downstream of the translational STOP codon of the calbindin 1 gene (*Daigle et al., 2018*). Further details of these lines can be found in the Key resources table. The following crosses were generated by cross-breeding these lines: *Trpm8*^Flp;RCE:FRT mice, which were used for anatomical analysis of Trpm8-expressing DRG cells, for electrophysiological experiments, and for retrograde tracing of spinothalamic neurons; *Trpm8*^Flp;RCE:FRT;Ai9 mice, which were used for some electrophysiological experiments; *Trpm8*^Flp;RCE:FRT;*Phox2a*::Cre;Ai9 mice, which were used for the combined confocal and electron microscopic analysis; and *Calb1*^Cre;*Trpm8*^Flp;RCE:FRT mice, which were used to determine the proportion of retrogradely labelled Calb1-expressing cells that received dense Trpm8 input.

Perfusion fixation was performed while mice were deeply anaesthetised with pentobarbitone (20 mg i.p.). A brief rinse with Ringer's solution was followed by perfusion with ~250 ml of fixative. In all cases, apart from those used for combined confocal/electron microscopy, the fixative consisted of 4% freshly depolymerised formaldehyde in phosphate buffer (PB). Spinal cords and brains were dissected and stored in the same fixative for 2 hr, before being transferred to 30% sucrose in PB.

Dorsal root ganglia for FISH were obtained from *Trpm8*^Flp;RCE:FRT mice, which had been killed with an overdose of pentobarbital (40 mg i.p.) and perfused with cold phosphate-buffered saline (PBS). The ganglia were fixed in 4% formaldehyde for 2 hr.

Mice of both sexes, aged 4–14 weeks and weighing between 13 and 25 g at the start of any surgical procedures were used in the study.

### Fluorescent in situ hybridisation, immunohistochemistry, and imaging

FISH was performed on intact lumbar dorsal root ganglia from *Trpm8*^Flp;RCE:FRT mice using HCR RNA-FISH v3.0 reagents. Tissues were dehydrated in graded ethanol solutions, followed by rinsing in PBS. Hybridisation and amplification were performed with a probe against mouse *Trpm8* mRNA and hairpins conjugated to Alexa Fluor 546 (HCR v3.0, Molecular Instruments, Inc) according to the manufacturer's instructions. Tissues were mounted using antifade mounting medium and stored at –20 °C.

Sections of brain, spinal cord, and dorsal root ganglia for immunohistochemical processing were cut on either a vibrating blade microtome (Leica VT1000 or VT1200) or a cryostat (Leica CM1950). Section thickness was 50 µm for brain and 60 µm for spinal cord and dorsal root ganglia. Multiple-label immunofluorescence microscopy was carried out as described previously (*Ma et al., 2025b*; *Polgár et al., 2023*; *Quillet et al., 2023*). Reactions were performed on free-floating sections, and the sources and concentrations of antibodies are listed in the Key resources table. Sections were incubated for 1–3 days at 4 °C or room temperature in primary antibodies diluted in PBS that contained

0.3 M NaCl, 0.3% Triton X-100 (except for tissue processed by the confocal/EM method), and 5% normal donkey serum. After rinsing, they were incubated for 2–18 hr in appropriate species-specific secondary antibodies that were raised in donkey and conjugated to Alexa Fluor Plus 405, Alexa Fluor 488, Alexa Fluor Plus 488, Alexa Fluor Plus 555, Alexa Fluor 647, or Rhodamine Red, in the same diluent. Following the reaction, sections were mounted with anti-fade medium and stored at –20 °C.

Tissue sections and dorsal root ganglia that had undergone immunofluorescence labelling or FISH were scanned with either a Zeiss LSM710 confocal microscope with Argon multi-line, 405 nm diode, 561 nm solid state, and 633 nm HeNe lasers, or with a Zeiss LSM900 Airyscan confocal microscope with 405, 488, 561, and 640 nm diode lasers. Confocal image stacks were obtained through a 20 x dry lens (numerical aperture, NA, 0.8), or through 40 x (NA 1.3), or 63 x (NA 1.4) oil immersion lenses with the confocal aperture set to 1 Airy unit or less.

The method for combining confocal and electron microscopy was the same as that described previously (*Bell et al., 2020*). Two *Trpm8*$^{Flp}$;RCE:FRT;*Phox2a*::Cre;Ai9 mice were perfused with fixative that contained 0.5% glutaraldehyde and 4% formaldehyde. Horizontal sections of spinal cord were obtained from lumbar segments, and these were treated with 50% ethanol for 30 min to enhance antibody penetration, and then with 1% sodium borohydride for 30 min to quench unbound aldehyde sites. After rinsing, they were incubated for 3 days in primary antibodies against GFP (raised in rabbit) and mCherry (raised in rat), and then overnight in a mixture of secondary antibodies: donkey anti-rabbit conjugated to Alexa Fluor 488, donkey anti-rabbit conjugated to biotin, and donkey anti-rat conjugated to Rhodamine Red. Note that the anti-mCherry antibody recognises tdTomato. The antibody diluent consisted of PBS that contained 0.3 M NaCl and 5% normal donkey serum. They were then incubated in avidin conjugated to horseradish peroxidase (HRP) and mounted on slides with anti-fade medium. Sections were viewed with a confocal microscope, and regions that contained tdTomato-positive cells that were densely coated with GFP-labelled axons were selected. These were scanned with the confocal microscope to generate z-series through the cell bodies and dendritic trees of the neurons. The sections were then removed from the slides, rinsed in PBS and then PB, processed with 3,3'-diaminobenzidine (DAB) and $H_2O_2$ to reveal peroxidase-labelled (Trpm8-positive) axons, osmicated, stained with uranyl acetate, dehydrated, and flat-embedded in Durcupan resin between acetate sheets (*Bell et al., 2020*). Series of ultrathin sections (silver interference colour) were cut with a diamond knife and collected on Formvar-coated slot grids. These were further stained with lead citrate and viewed with either a Philips CM100 or Jeol JEM-1400Flash electron microscope.

## Intracranial and intraspinal injections

All surgical procedures involving intraspinal and/or intracranial injections were performed while the mice were anaesthetised with isoflurane (1–2%), and all animals received perioperative analgesia (buprenorphine 0.1 mg/kg and carprofen 10 mg/kg, s.c.). For both types of injection, the mice were placed in a stereotaxic frame.

For intracranial injections, lidocaine (10 mg/kg, s.c.) was injected into the skin over the skull, which was then incised. Burr holes were drilled to allow targeting at the following co-ordinates (CVLM: 1.1–1.3 mm lateral to midline, 3.5 mm caudal to lambdoid suture, 3.3–3.7 mm below brain surface; LPB: 1.1 mm lateral to the midline, 0.35 mm caudal to lambdoid suture or 4.5 mm caudal to bregma, 3.0 mm below brain surface, PAG: 0.4 mm lateral to midline, 0.4 mm caudal to the interaural line, 3.1 mm below brain surface; VPL: 1.6–1.7 mm lateral to the midline, 0.8–1.1 mm caudal to bregma, 3.4 mm below brain surface). Injections were made through glass micropipettes (outer tip diameter ~60 μm) attached to a Harvard Pump 11 Elite (for AAV injections) or a PV 800 Pneumatic Picopump (WPI; for CTB injections). For optogenetic experiments, *Trpm8*$^{Flp}$RCE:FRT mice that had received intraperitoneal injections of AAV.PHP.S.Flp$^{ON}$.ChR2_YFP (see below) were injected with AAV11.tdTomato (2.5×10$^9$ GC in 500 nl) into the LPB or CVLM. Brain injections for the electrophysiological experiments in which the semi-intact preparation was used to record from cells with or without dense Trpm8 input consisted of AAV9.mCherry (7.5×10$^8$–1.2×10$^{10}$ GC in 500 nl) injected into the LPB or CVLM of *Trpm8*$^{Flp}$;RCE:FRT mice, or AAV9.Cre_GFP (3.5×10$^9$ GC in 500 nl) injected into the CVLM of a *Trpm8*$^{Flp}$;RCE:FRT;Ai9 mouse. For anatomical experiments involving *Calb1*$^{Cre}$;*Trpm8*$^{Flp}$;RCE:FRT mice AAV11.Cre$^{ON}$.tdTomato was injected into LPB (2×10$^9$ GC in 400 nl) or PAG (1×10$^9$ GC in 200 nl). For the immunohistochemical analysis of calbindin expression in Trpm8-innervated neurons, *Trpm8*$^{Flp}$;RCE:FRT mice received injections of AAV9.mCherry (7.5×10$^8$–1.2×10$^{10}$ GC in 500 nl) into the

LPB. For electrophysiological recording of Calb1-expressing neurons in the semi-intact preparation, 4 *Calb1*[Cre] mice were injected with AAV11.Cre[ON].tdTomato into the CVLM (2×10⁹ GC in 400 nl). Injections into the lateral thalamus consisted of 100 nl of 1% CTB. Micropipettes were left in place for ~5 mins after the injection to limit leakage up the track. For electrophysiological experiments, mice were allowed to recover for between 9 and 22 days (see below for further details). For anatomical experiments, survival times were 3 days for CTB injections and 12–16 days for AAV injections and these were followed by perfusion fixation, as described above.

For intraspinal injections in lumbar spinal cord, the T12 and L1 vertebrae were exposed and clamped. Injections were made through glass micropipettes (outer tip diameter ~60 μm) attached to a 10 μL Hamilton syringe, on the right side of the spinal cord into either the L3 segment, or the L3, L4, and L5 segments. Injections into L3 and L5 were made through the intervertebral spaces on either side of the T13 vertebra, while those into L4 were made through a small hole drilled in the lamina of the T13 vertebra. In initial experiments, we used a similar strategy to that described in previous papers (*Ma et al., 2025a*; *Ma et al., 2025b*), and these are referred to as 'central' injections. We subsequently modified the technique in order to restrict injection sites to the medial part of the dorsal horn and exclude the LSN, and we refer to these as 'medial' injections. For cervical injections, the head was held in place with ear-bars and the T2 spine was clamped. Injections were made (as described above) in the intervertebral space between the C6 and C7 vertebrae. The mice survived between 26 and 57 days and were then deeply anaesthetised and perfused with fixative as described above (for further details see *Table 1*).

## Image analysis

Lamina I ALS neurons that received numerous contacts from GFP-positive (Trpm8-expressing) primary afferents were identified in several parts of this study, either following retrograde labelling or through expression of tdTomato (in crosses that involved *Phox2a*::Cre and Ai9). The cells could be readily identified by the close association of GFP-labelled axons with their dendrites, which received multiple contacts from GFP-positive boutons. In most cases, the cell bodies were also associated with numerous GFP-labelled boutons, and these sometimes completely surrounded the cell body (*Bell et al., 2024*).

All quantitative image analyses were performed with Neurolucida for Confocal software (MBF Bioscience, Williston, VT, USA).

To determine the proportion of DRG neurons that were labelled with GFP in *Trpm8*[Flp];RCE:FRT mice, we scanned three or four sections immunostained to reveal GFP and NeuN, and counterstained with DAPI from the L4 dorsal root ganglia from each of three mice (one male, two female) with a confocal microscope. Z-stacks that included the complete cross-sectional area of the ganglion were analysed using a modified disector method (*Polgár et al., 2004*). Reference and look-up sections were set 10 μm apart, and all intervening sections were examined to reveal any neurons that might have been located between these two levels. Initially, we viewed the NeuN and DAPI channels, and in this way, we identified all neurons (between 904 and 1383 per mouse) for which the top surface of the nucleus was located between reference and look-up section. We then revealed the channel for GFP and determined which of the sampled neurons that were GFP-immunoreactive.

For the comparison of *Trpm8* mRNA and GFP, dorsal root ganglia from 3 *Trpm8*[Flp];RCE:FRT mice (one male, two female) that had been reacted as described above were scanned with a confocal microscope to generate z-series through the entire volume of the ganglion. Initially, both fluorescent channels (GFP and Alexa Fluor 546) were combined such that any cell with either GFP or *Trpm8* mRNA was visible. For all of these cells (between 155–209 per mouse), the cell size was determined by drawing an outline of the cell at its maximum point and converting cross-sectional area to diameter by assuming a circular shape. The two fluorescent channels were then separated, and for each cell, the presence of GFP and/or *Trpm8* mRNA was determined.

Because we found a relatively limited range of sizes for GFP/*Trpm8*-positive cells (see *Figure 1C*), we did not use a stereological method for analysing co-localisation with other neurochemical markers. We scanned sections of dorsal root ganglia immunoreacted to reveal GFP together with: (1) VGLUT3, (2) Trpv1, (3) CGRP and substance P, or (4) somatostatin. In each case, tissue from three mice (either two male and one female, or one male and two female) was used, and z-stacks covering the entire cross-sectional area of the ganglion were analysed. Initially, the GFP channel was viewed and the

locations of all labelled cells were marked (between 126 and 350 per mouse for each reaction). The other channel(s) was/were then examined and the presence or absence of the other marker(s) was noted.

To determine the proportion of retrogradely labelled lamina I neurons that had dense input from Trpm8-expressing afferents in the *Calb1*^Cre^;*Trpm8*^Flp^;RCE:FRT mice injected with AAV11.Cre^ON^.tdTomato into either LPB (four mice, two male, and two female) or PAG (three mice, one male, two female), we examined horizontal sections cut through the L2, L3, and L4 segments. These had been immunostained to reveal GFP and tdTomato, and were scanned with a confocal microscope. The resulting z-stacks were viewed with Neurolucida software, initially with only the tdTomato channel visible, and the locations of retrogradely labelled lamina I neurons were recorded (between 32 and 81 cells per segment in each animal for the experiments involving LPB injections). For the experiments involving PAG injections, numbers of retrogradely labelled neurons were much lower, and results from the three segments were pooled for each animal (between 26 and 99 cells per mouse). The GFP channel was then viewed, and cells that received dense Trpm8 input were identified. The L5 segments of the 4 *Calb1*^Cre^;*Trpm8*^Flp^;RCE:FRT mice that had received injections into LPB were cut transversely and the locations of tdTomato-labelled cell bodies on between 8 and 10 individual sections were identified. These were then morphed onto a standard outline of the L5 segment in Inkscape, resulting in single scalable vector graphics files for each mouse.

To test whether cold-selective cells were included among those projecting to the lateral thalamus, we examined horizontal sections from the L4 and C7 segments of 3 *Trpm8*^Flp^;RCE:FRT mice (two male, one female) that had received CTB injections into the lateral thalamus. Sections were immunoreacted to reveal CTB and GFP and scanned with a confocal microscope. We then searched for CTB-labelled cells that were associated with numerous GFP-positive axons.

For anterograde tracing following intraspinal injections of AAV1.Cre^ON^.tdTomato in *Calb1*^Cre^ mice, sections were first viewed with a fluorescence microscope (Zeiss Axioscope 5) and selected sections were scanned with a confocal microscope. Identification of brain regions that contained tdTomato-labelled axons was generally based on the atlas of *Franklin and Paxinos, 2007*, apart from the LPB. For this, we used the same terminology as in our previous paper (*Ma et al., 2025a*), and identified the PBrel nucleus based on the findings of *Geerling et al., 2016*.

## Whole spinal cord preparation

The whole spinal cord preparation was made as described previously with minor modifications (*Hachisuka et al., 2018*). Animals were anaesthetised with isoflurane, followed by a lethal dose of pentobarbital, and then quickly perfused with oxygenated sucrose-based artificial cerebrospinal fluid (ACSF; in mM; 3.0 KCl, 1.2 NaH2PO4, 0.5 CaCl2, 7.0 MgCl2, 26.0 NaHCO3, 15.0 glucose, 251.6 sucrose, 1.0 Na ascorbate, and 1.0 Na pyruvate). Immediately after perfusion, the back skin was incised, and the spinal cord was quickly excised and placed into a sucrose-based ACSF. Dura and pia-arachnoid membranes were removed. The spinal cord was pinned into a chamber wall made from Sylgard. The spinal cord was perfused with oxygenated normal ACSF (in mM; 125.8 NaCl, 3.0 KCl, 1.2 $NaH_2PO_4$, 2.4 $CaCl_2$, 1.3 $MgCl_2$, 26.0 $NaHCO_3$, and 15.0 glucose) at 25 °C.

## Semi-intact somatosensory preparation

The semi-intact somatosensory preparation was made as described previously (*Hachisuka et al., 2016*; *Hachisuka et al., 2020*) with minor modifications. Mice were anaesthetised with isoflurane, followed by a lethal dose of pentobarbital, and then quickly perfused with oxygenated sucrose-based artificial cerebrospinal fluid (ACSF; in mM; 3.0 KCl, 1.2 $NaH_2PO_4$, 0.5 $CaCl_2$, 7.0 $MgCl_2$, 26.0 $NaHCO_3$, 15.0 glucose, 251.6 sucrose, 1.0 Na ascorbate, and 1.0 Na pyruvate). The back skin was incised, and the spinal cord was exposed by dorsal laminectomy. The right hindlimb and the right side of the trunk with the spinal cord were excised and transferred to the dissection/recording chamber, and tissues were submerged in the oxygenated sucrose-based ACSF. The right hind paw skin, saphenous nerve, and femoral cutaneous nerve were carefully isolated from the surrounding tissues. Only the L2 and L3 ganglia on the right side were left attached to the spine. All dorsal roots except L2 and L3, dural and pial membranes were carefully removed. The spinal cord was pinned onto a Sylgard chamber with the right dorsal horn facing upward. The tissue was perfused with oxygenated normal ACSF (in mM; 125.8 NaCl, 3.0 KCl, 1.2 $NaH_2PO_4$, 2.4 $CaCl_2$, 1.3 $MgCl_2$, 26.0 $NaHCO_3$, and 15.0 glucose)

at 25 °C. Recordings were performed for up to 4 hr after dissection. The hindpaw skin was also at a temperature of 25 °C. At this temperature, Trpm8 afferents will have been active, but are likely to have adapted over the course of the experiment. In addition, the conduction velocity of their axons will have been lower than in the in vivo state.

## Patch-clamp recording from lamina I projection neurons

Neurons were visualised using a fixed stage upright microscope (SliceScope; Scientifica, Uckfield, UK) equipped with a 40 x water immersion objective and a CMOS Camera (Prime BSI Express, Teledyne Vision Solutions, Ontario, Canada). A narrow-beam infrared LED (Opto SFH 4550; Osram, Munich, Germany, emission peak, 860 nm) was positioned outside the solution meniscus. Fluorescent cells and axons were visualised using LED illumination (pE-300 ultra, CoolLED, Andover, UK). Whole-cell patch clamp recordings were made with a pipette constructed from thin-walled single-filamented borosilicate glass using a microelectrode puller (P-1000, Sutter Instrument, CA, USA). Pipette resistances ranged from 6 to 10 M$\Omega$. Electrodes were filled with an intracellular solution (in mM; 130.0 K gluconate, 10.0 KCl, 2.0 MgCl$_2$, 10.0 HEPES, 0.5 EGTA, 2.0 ATP-Na, 0.5 GTP-Na, and 0.2% Neurobiotin, pH adjusted to 7.3 with 1.0 M KOH). Data were recorded and acquired with an amplifier (Axopatch 200B; Molecular Devices, Wokingham, UK). The data were low-pass filtered at 2 kHz and digitised at 10 kHz with an A/D converter (Digidata 1550B; Molecular Devices, Wokingham, UK) and stored using a data acquisition program (Clampex 11; Molecular Devices, Wokingham, UK).

## Optogenetic activation

Two *Trpm8*[Flp];RCE:FRT mice were given intraperitoneal injections of AAV.PHP.S.Flp[ON].ChR2_YFP (1.3×10[11] GC in 10 µl) on postnatal day 2, and 5 weeks later they received brain injections of AAV11. tdTomato, as described above. A blue light pulse (pE-300 ultra, CoolLED, Andover, UK) was applied through the objective (40 x) of the microscope for 1 ms during whole-cell patch-clamp recordings from tdTomato-labelled lamina I neurons in the whole-cord preparation. The shutter was controlled by a TTL pulse from the A/D converter (Digidata 1550B; Molecular Devices, Wokingham, UK). To determine whether the recorded neuron received monosynaptic or polysynaptic input from Trpm8 afferents, we applied 0.2 Hz photostimulation (1 ms pulse width) based on a previous study (*Hachisuka et al., 2018*) with minor modifications. Input was considered monosynaptic if there was no failure and the latency jitter was smaller than 1 ms.

## Natural stimulation to the skin

To test whether cells with dense Trpm8 input were cold-selective, we made patch-clamp recordings in mice in which Trpm8 axons were revealed with GFP and projection neurons were labelled with either mCherry or tdTomato using the semi-intact somatosensory preparation (*Figure 4A*). Successful recordings were made in 6 out of 16 experiments: 5 *Trpm8*[Flp];RCE:FRT mice (two male, three female) that had received injections of AAV9.mCherry into the LPB or CVLM, and one male *Trpm8*[Flp];RCE:FRT;Ai9 mouse injected in the CVLM with AAV9.EGFP_Cre. For the investigation of *Calb1*-Cre-positive lamina I neurons, recordings were made in four female *Calb1*[Cre] mice that had received injections of AAV11.Cre[ON].tdTomato into the CVLM.

Recordings were initially made in voltage clamp mode (VH = –70 mV) to reveal spontaneous EPSCs. To find the receptive field of the recorded neurons, mechanical, hot, and cold stimuli were gently applied to the skin. Mechanical stimulation was applied using von Frey filaments (4 g or 10 g), and in some cases, the skin was brushed with a paintbrush. Hot and cold stimulation were administered by applying a small amount of 15 °C saline or 50 °C saline to the skin using an eye dropper. The temperature at the skin surface was 20°C and 38°C, respectively. Once the receptive field was identified, each stimulus was reapplied directly to the receptive field for 1 s. The evoked EPSCs were recorded in voltage clamp mode (VH = –70 mV), and the evoked action potentials were recorded in current clamp mode (IH = 0 pA) or both. The events were detected with Easy Electrophysiology software (Easy Electrophysiology, London, UK).

## Quantification and statistical analysis

Data are reported as mean ± SD, unless stated otherwise. Statistical analyses were performed using Prism software (v10, GraphPad Software, CA, USA) or SciPy library (https://docs.scipy.org/doc/) in

Python. The statistical tests used for each experiment, including tests for multiple comparisons, are given in the appropriate figure legends. A p-value of <0.05 was considered significant, and significance markers are denoted within figures as follows: $*p<0.05$, $**p<0.01$, $***p<0.001$.

## Acknowledgements

This research was funded in whole, or in part, by the Wellcome Trust (Grant numbers 219433/Z/19/Z and 304005/Z/23/Z) and the Medical Research Council (Grant number MR/V033638/1). For the purpose of Open Access, the authors have applied a CC BY public copyright licence to any Author Accepted Manuscript version arising from this submission. We are grateful to Robert Kerr, Iain Plenderleith, and Erin Dunn for expert technical assistance, and to Mark Hoon and Artur Kania for the gifts of *Trpm8*[Flp] and *Phox2a*::Cre mice.

## Additional information

### Funding

| Funder | Grant reference number | Author |
|---|---|---|
| Wellcome Trust | 10.35802/219433 | Andrew J Todd |
| Medical Research Council | MR/V033638/1 | Junichi Hachisuka |
| Wellcome Trust | 10.35802/304005 | Andrew M Bell |

The funders had no role in study design, data collection and interpretation, or the decision to submit the work for publication. For the purpose of Open Access, the authors have applied a CC BY public copyright license to any Author Accepted Manuscript version arising from this submission.

### Author contributions

Aimi N Razlan, Wenhui Ma, Conceptualization, Data curation, Formal analysis, Investigation, Visualization, Writing – original draft, Writing – review and editing; Allen C Dickie, Conceptualization, Data curation, Investigation, Writing – original draft; Erika Polgar, Conceptualization, Investigation, Writing – original draft; Anna G McFarlane, Data curation, Formal analysis, Investigation, Visualization, Writing – original draft; Mansi Yadav, Formal analysis, Investigation, Writing – original draft; Andrew H Cooper, Douglas Strathdee, Investigation, Writing – original draft; Masahiko Watanabe, Resources, Writing – original draft; Andrew M Bell, Conceptualization, Supervision, Funding acquisition, Writing – original draft, Writing – review and editing; Andrew J Todd, Conceptualization, Data curation, Formal analysis, Supervision, Funding acquisition, Investigation, Visualization, Methodology, Writing – original draft, Project administration, Writing – review and editing; Junichi Hachisuka, Conceptualization, Formal analysis, Supervision, Funding acquisition, Investigation, Visualization, Methodology, Writing – original draft, Project administration, Writing – review and editing

### Author ORCIDs

Aimi N Razlan ⓘ https://orcid.org/0000-0003-0703-760X
Wenhui Ma ⓘ https://orcid.org/0009-0003-5489-4396
Allen C Dickie ⓘ https://orcid.org/0000-0002-6339-2801
Andrew H Cooper ⓘ https://orcid.org/0000-0003-4737-9364
Douglas Strathdee ⓘ https://orcid.org/0000-0003-2959-4327
Masahiko Watanabe ⓘ https://orcid.org/0000-0001-5037-7138
Andrew M Bell ⓘ https://orcid.org/0000-0001-6510-4423
Andrew J Todd ⓘ https://orcid.org/0000-0002-3007-6749
Junichi Hachisuka ⓘ https://orcid.org/0000-0003-3987-4381

### Ethics

All experiments were approved by the Animal Welfare and Ethical Review Body of the University of Glasgow, and were carried out in accordance with the UK Animals (Scientific Procedures) Act 1986 and ARRIVE guidelines.

Reviewer #1 (Public review): https://doi.org/10.7554/eLife.109502.4.sa1
Reviewer #2 (Public review): https://doi.org/10.7554/eLife.109502.4.sa2
Reviewer #3 (Public review): https://doi.org/10.7554/eLife.109502.4.sa3
Author response https://doi.org/10.7554/eLife.109502.4.sa4

# Additional files

**Supplementary files**
MDAR checklist

**Data availability**
The datasets generated and analysed during the current study can be accessed from the University of Glasgow's Enlighten repository https://doi.org/10.5525/gla.researchdata.2235. This study did not generate any new materials, reagents or code.

The following dataset was generated:

| Author(s) | Year | Dataset title | Dataset URL | Database and Identifier |
|---|---|---|---|---|
| Razlan AN, Ma W, Dickie A, Polgár E, McFarlane A, Yadav M, Cooper A, Strathdee D, Watanabe M, Bell A, Todd A, Hachisuka J | 2026 | Characterisation of cold-selective lamina I spinal projection neurons | https://doi.org/10.5525/gla.researchdata.2235 | Enlighten Research Data, 10.5525/gla.researchdata.2235 |

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

# Appendix 1

**Appendix 1—key resources table**

| Reagent type (species) or resource | Designation | Source or reference | Identifiers | Additional information |
|---|---|---|---|---|
| Strain, strain background (mouse) | *Trpm8*$^{Flp}$ | Dr Mark Hoon | | |
| Strain, strain background (mouse) | B6.Cg-Gt(ROSA)26Sor$^{tm9(CAG-tdTomato)Hze}$/J (Ai9) | The Jackson Laboratory | Cat#: 007909 RRID:IMSR_JAX:007909 | |
| Strain, strain background (mouse) | *Phox2a*::Cre | Dr Artur Kania | | |
| Strain, strain background (mouse) | STOCK *Gt(ROSA)26Sor*$^{tm1.2(CAG-EGFP)Fsh}$/Mmjax (RCE:FRT) | The Jackson Laboratory | Cat#: 032038 RRID:MMRRC_032038-JAX | |
| Strain, strain background (mouse) | B6;129S-Calb1tm2.1(cre)Hze/J (Calb1Cre) | The Jackson Laboratory | Cat#: 028532 RRID:IMSR_JAX:028532 | |
| Transfected construct (AAV) | pENN.AAV.CB7.CI.mCherry.WPRE.RBG (AAV9.mCherry) | Addgene | Cat#: 105544-AAV9 RRID:Addgene_105544 | |
| Transfected construct (AAV) | pssAAV-2-CAG-EGFP_Cre-WPRE-SV40p(A) (AAV9.EGFP_Cre) | VVF, Zurich | Cat#: v25-9 | |
| Transfected construct (AAV) | pssAAV-2-CAG-dlox-tdTomato(rev)-dlox-WPRE-bGHp(A) (AAV1.Cre$^{ON}$.tdTomato) | VVF, Zurich | Cat#: v167-1 | |
| Transfected construct (AAV) | AAV.PHP.S.Flp$^{ON}$.ChR2_YFP | VVF, Zurich | Cat#: v237-PHP.S | |
| Transfected construct (AAV) | AAV11.CAG.Cre$^{ON}$.tdTomato (AAV11.Cre$^{ON}$.tdTomato) | BrainCase Biotechnology Co Ltd, Wuhan, China | Cat#: BC-0870 | |
| Transfected construct (AAV) | AAV11.CAG.tdTomato (AAV11.tdTomato) | BrainCase Biotechnology Co Ltd, Wuhan, China | Cat#: BC-0868 | |
| Antibody | anti-GFP (Chicken polyclonal) | Abcam | Cat#: ab13970 RRID:AB_300798 | (1:1000) |
| Antibody | anti-GFP (Rabbit polyclonal) | M Watanabe | RRID:AB_2571573 | (1:1000) |
| Antibody | anti-mCherry (Rat polyclonal) | Thermo Fisher Scientific | Cat#: M11217 RRID:AB_2536611 | (1:1000) |
| Antibody | anti-NeuN (Guinea pig polyclonal) | Synaptic Systems | Cat#: 266 004 RRID:AB_2619988 | (1:1000) |
| Antibody | anti-NeuN-Alexa Fluor 647 (Rabbit monoclonal) | Abcam | Cat#: EPR12763 RRID:AB_2532109 | (1:1000) |
| Antibody | anti-substance P (Rat monoclonal) | Oxford Biotechnology | Cat#: OBT06435 | (1:200) |
| Antibody | anti-substance P (Rabbit polyclonal) | Peninsula | Cat#: T-4107 RRID:AB_518630 | (1:1000) |
| Antibody | anti-TRPV1 (Rabbit polyclonal) | Synaptic Systems | Cat#: 444 013 RRID:AB_2864792 | (1:1000) |
| Antibody | anti-CGRP (Sheep polyclonal) | Enzo | Cat#: BML-CA1137 RRID:AB_2243859 | (1:2000) |
| Antibody | anti-somatostatin (Rabbit polyclonal) | Peninsula | Cat#: T-4103 RRID:AB_518614 | (1:500) |

*Appendix 1 Continued on next page*

*Appendix 1 Continued*

| Reagent type (species) or resource | Designation | Source or reference | Identifiers | Additional information |
|---|---|---|---|---|
| Antibody | anti-VGLUT3 (Guinea pig polyclonal) | M Watanabe | RRID:AB_2571856 | (1:100) |
| Antibody | anti-Foxp2 (Sheep polyclonal) | Biotechne | Cat#: AF5647 RRID:AB_2107133 | (1:500) |
| Antibody | anti-cholera toxin B subunit (Goat polyclonal) | List Biological | Cat#: 703 RRID:AB_10013220 | IF (1:1000) IP (1:100,000) |
| Antibody | anti-Homer1 (Goat polyclonal) | M Watanabe | RRID:AB_2631104 | (1:500) |
| Antibody | anti-calbindin (Rabbit polyclonal) | Swant | Cat#: CB38 RRID:AB_10000340 | (1:1000) |
| Antibody | Alexa Fluor Plus 405 anti-Rabbit IgG (Donkey polyclonal) | Thermo-Fisher Scientific | Cat#: A48258 RRID:AB_2890547 | (1:500) |
| Antibody | Alexa Fluor 488 Anti-Chicken IgY (Donkey polyclonal) | Jackson ImmunoResearch | Cat#: 703-545-155 RRID:AB_2340375 | (1:500) |
| Antibody | Alexa Fluor 488 anti-Goat IgG (Donkey polyclonal) | Jackson ImmunoResearch | Cat#: 705-545-147 RRID:AB_2336933 | (1:500) |
| Antibody | Alexa Fluor 488 anti-Goat IgG (Donkey polyclonal) | Jackson ImmunoResearch | Cat#: 705-545-147 RRID:AB_2336933 | (1:500) |
| Antibody | Alexa Fluor 488 Anti-Rabbit IgG (Donkey polyclonal) | Jackson ImmunoResearch | Cat#: 711-545-152 RRID:AB_2313584 | (1:500) |
| Antibody | Alexa Fluor Plus 488 anti-Rabbit IgG (Donkey polyclonal) | Thermo-Fisher Scientific | Cat#: A32790 RRID:AB_2762833 | (1:500) |
| Antibody | Rhodamine Red-X Anti-Guinea pig IgG (Donkey polyclonal) | Jackson ImmunoResearch | Cat#: 706-295-148 RRID:AB_2340468 | (1:100) |
| Antibody | Rhodamine Red-X Anti-Rabbit IgG (Donkey polyclonal) | Jackson ImmunoResearch | Cat#: 711-295-152 RRID:AB_2340613 | (1:100) |
| Antibody | Rhodamine Red-X Anti-Rat IgG (Donkey polyclonal) | Jackson ImmunoResearch | Cat#: 712-295-153 RRID:AB_2340676 | (1:100) |
| Antibody | Alexa Fluor Plus 555 anti-Rat IgG (Donkey polyclonal) | Thermo-Fisher Scientific | Cat#: A48270 RRID:AB_2896336 | (1:500) |
| Antibody | Alexa Fluor Plus 555 anti-Goat IgG (Donkey polyclonal) | Thermo-Fisher Scientific | Cat#: A32816 RRID:AB_2762839 | (1:500) |
| Antibody | Alexa Fluor 647 Anti-Rabbit IgG (Donkey polyclonal) | Jackson ImmunoResearch | Cat#: 711-605-152 RRID:AB_2492288 | (1:500) |
| Antibody | Alexa Fluor 647 Anti-Goat IgG (Donkey polyclonal) | Jackson ImmunoResearch | Cat#: 705-605-147 RRID:AB_2340437 | (1:500) |
| Antibody | Alexa Fluor 647 Anti-Guinea pig IgG (Donkey polyclonal) | Jackson ImmunoResearch | Cat#: 706-605-148 RRID:AB_2340476 | (1:500) |
| Antibody | Biotin-SP Anti-Goat IgG (Donkey polyclonal) | Jackson ImmunoResearch | Cat#: 705-065-147 RRID:AB_2340397 | (1:500) |
| Antibody | Biotin-SP Anti-Guinea pig IgG (Donkey polyclonal) | Jackson ImmunoResearch | Cat#: 706-065-148 RRID:AB_2340451 | (1:500) |
| Antibody | Biotin-SP Anti-Rabbit IgG (Donkey polyclonal) | Jackson ImmunoResearch | Cat#: 711-065-152 RRID:AB_2340593 | (1:500) |
| Sequence-based reagent | Mm-Trpm8 v3.0 (HCR probe) | Molecular Instruments | | |
| Peptide, recombinant protein | Extravidin-Peroxidase | Sigma-Aldrich | Cat#: E2886 RRID:AB_2620165 | |
| Chemical compound, drug | Extravidin-Peroxidase | Sigma-Aldrich | Cat#: E2886 RRID:AB_2620165 | |
| Chemical compound, drug | Streptavidin-Pacific Blue | Life Technologies | Cat#: S11222 | |

*Appendix 1 Continued on next page*

*Appendix 1 Continued*

| Reagent type (species) or resource | Designation | Source or reference | Identifiers | Additional information |
|---|---|---|---|---|
| Chemical compound, drug | Cholera toxin B subunit | Sigma-Aldrich | Cat#: C9972 | |
| Chemical compound, drug | DAPI | Sigma-Aldrich | Cat#: D9542 | |
| Chemical compound, drug | Sodium borohydride | Sigma-Aldrich | Cat#: 452882 | |
| Chemical compound, drug | 3,3′-Diaminobenzidine | Revvity | Cat#: NEL938001EA | |
| Chemical compound, drug | Hydrogen peroxide | Sigma-Aldrich | Cat#: H1009 | |
| Chemical compound, drug | Osmium tetroxide | Agar Scientific | Cat#: R1016 | |
| Chemical compound, drug | Uranyl acetate | Agar Scientific | Cat#: R1260A | |
| Chemical compound, drug | Lead citrate | Agar Scientific | Cat#: R1210 | |
| Software, algorithm | Neurolucida | MBF Bioscience | https://www.mbfbioscience.com/neurolucida RRID:SCR_001775 | |
| Software, algorithm | Neurolucida Explorer | MBF Bioscience | https://www.mbfbioscience.com/neurolucida-explorer RRID:SCR_017348 | |
| Software, algorithm | pClamp | Molecular Devices | https://www.moleculardevices.com/products/axon-patch-clamp-system/acquisition-and-analysis-software/pclamp-software-suite#gref RRID:SCR_011323 | |
| Software, algorithm | Easy Electrophysiology | Easy Electrophysiology LTD | https://www.easyelectrophysiology.com | |
| Software, algorithm | Zen Black | Carl Zeiss | https://www.zeiss.com/microscopy/int/products/microscope-software/zen.html RRID:SCR_018163 | |
| Software, algorithm | Prism | GraphPad Software | https://www.graphpad.com/scientific-software/prism/ RRID:SCR_002798 | |
| Software, algorithm | Photoshop CS6 | Adobe Systems Incorporated | https://www.adobe.com/ | |
| Software, algorithm | Xara Xtreme v2.0 | Xara Group Ltd | https://www.xara.com/ | |
| Software, algorithm | Inkscape | Software Freedom Conservancy | https://inkscape.org/ | |

