## [Editor Report · eLife Assessment]

This **important** study offers insights into the anatomical and physiological features of cold-selective lamina I spinal projection neurons. The evidence supporting the authors' claims is **convincing**, although including a larger sample size and more quantification would have strengthened the study, and the claims of monosynaptic connectivity would benefit from further experimental evidence. The work will interest those in the field of somatosensory biology, especially researchers studying spinal cord dorsal horn circuits and projection neuron cell types

---

## [Referee Report · Reviewer #1 (Public review)]

[Editors' note: this version has been assessed by the Reviewing Editor without further input from the original reviewers.]

Summary:

Spinal projection neurons in the anterolateral tract transmit diverse somatosensory signals to the brain, including touch, temperature, itch, and pain. This group of spinal projection neurons is heterogeneous in their molecular identities, projection targets in the brain, and response properties. While most anterolateral tract projection neurons are multimodal (responding to more than one somatosensory modality), it has been shown that cold-selective projection neurons exist in lamina I of the spinal cord dorsal horn. Using a combination of anatomical and physiological approaches, the authors discovered that the cold-selective lamina I projection neurons are heavily innervated by Trpm8+ sensory neuron axons, with calb1+ spinal projection neurons primarily capturing these cold-selective lamina I projection neurons. These neurons project to specific brain targets, including the PBNrel and cPAG. This study adds to the ongoing effort in the field to identify and characterize spinal projection neuron subtypes, their physiology, and functions.

Strengths:

(1) The combination of anatomical and physiological analyses is powerful and offers a comprehensive understanding of the cold-selective lamina I projection neurons in the spinal cord dorsal horn. For example, the authors used detailed anatomical methods, including EM imaging of Trpm8+ axon terminals contacting the Phox2a+ lamina I projection neurons. Additionally, they recorded stimulus-evoked activity in Trpm8-recipient neurons, carefully selected by visual confirmation of tdTomato and GFP juxtaposition, which is technically challenging.

(2) This study identifies, for the first time, a molecular marker (calb1) that labels cold-selective lamina I projection neurons. Although calb1+ projection neurons are not entirely specific to cold-selective neurons, using an intersectional strategy combined with other genes enriched in this ALS group or cold-induced FosTRAP may further enhance specificity in the future.

(3) This study shows that cold-selective lamina I projection neurons specifically innervate certain brain targets of the anterolateral tract, including the NTS, PBNrel, and cPAG. This connectivity provides insights into the role of these neurons in cold sensation, which will be an exciting area for future research.

Weaknesses:

(1) The sample size for the ex vivo electrophysiology conducted on the calb1+ lamina I projection neurons (Figure 5) is limited to a total of six recorded neurons. Given the difficulty and complexity of the preparation, this is understandable. Notably, since approximately 87% of lamina I projection neurons heavily innervated by Trpm8+ terminals are calb1+, these six recordings of such neurons in Figure 4E could also be calb1+.

---

## [Referee Report · Reviewer #2 (Public review)]

Summary:

In this study, the authors took advantage of a semi-intact ex vivo somatosensory preparation that includes hindlimb skin to characterize the response of projection neurons in the dorsal horn of the spinal cord to peripheral stimulation, including cold thermal stimuli. The main aim was to characterize the connectivity between peripheral afferents expressing the cold sensing receptor TRPM8 and a set of genetically tagged neurons of the anterolateral system (ALS). These ALS neurons expressed high levels of the calcium binding protein calbindin 1.

In addition, combining different viral tracing methods, the authors could identify the anatomical targets of this specific subset of projection neurons within the brainstem and diencephalon.

Strengths:

The use of a relatively new (seldom used previously) transgenic line to label TRPM8-expressing afferents, combined with the genetic characterization of a previously identified subset of projections neurons add specificity to the characterization. The transgenic line appears to capture well the subpopulation of Trpm8-expressing neurons.

In addition, the use of electron microscopy techniques makes the interpretation of the structural contacts more compelling

The writing is clear and the presentation of findings follows a logical flow.

Overall, this study provides solid, novel information about the brain circuits involved in cold thermosensation.

Weaknesses:

In the characterization of recorded neurons in close contact or in the absence of this contact with TRPM8 afferents, the number of recordedd neurons is relatively low. In addition, the strength of thermal stimuli is not very well controlled, preventing a more precise characterization of the connectivity.

The authors acknowledge that, technically, this is a very difficult preparation with very low yield as far as obtaining successful recordings. Moreover, the tissue needs to be maintained at room temperature which is obviously not ideal when characterizing cold thermoreceptors due to the unavoidable effects of low temperature on cold-activated receptors.

---

## [Referee Report · Reviewer #3 (Public review)]

Summary:

Razlan and colleagues provide a detailed anatomical characterization of lamina I projection neurons in the mouse spinal cord that are densely innervated by primary afferents activated by cooling of the skin. The authors validate a Trpm8-Flp mouse line, show synaptic contacts between Trpm8⁺ boutons and projection neurons at the ultrastructural level, and demonstrate at the physiological level that these neurons specifically respond to cooling stimuli. Next, by taking advantage of previous transcriptomic analysis of ALS neurons, the authors identify calbindin as a marker for cold activatetd lamina I projection neurons and map their ascending projections to the rostral lateral parabrachial area, caudal periaqueductal gray, and ventral posterolateral thalamus, well-known thermosensory and thermoregulatory centers. Altogether, these findings provide strong anatomical and functional evidence for a direct line of transmission from Trpm8⁺ sensory afferents through Calb1⁺ lamina I neurons to key supraspinal centers controlling perception of cold and thermoregulatory responses.

Strengths:

The combination of mouse genetics, electron microscopy, ex-vivo physiology, optogenetics and viral tracing provides convincing evidence for a direct cold pathway. The work validates the Trpm8-Flp line by extensive anatomical and molecular characterization. Integration with previous transcriptomic and anatomical data, neatly links the cold-selective lamina I neurons to a molecularly defined cluster of ALS neurons, strengthening the bridge between molecular identity, anatomy, and physiological function.

Weaknesses:

The main limitation remains the relatively small number of neurons that could be recorded electrophysiologically. While understandable given the complexity of the preparation, this necessarily limits generalization.

---

## [Author Response]

The following is the authors’ response to the previous reviews

**Public reviews:**

**Reviewer #1 (Public review):**
The sample size for the ex vivo electrophysiology conducted on the calb1+ lamina I projection neurons (Figure 5) is limited to a total of six recorded neurons. Given the difficulty and complexity of the preparation, this is understandable. Notably, since approximately 87% of lamina I projection neurons heavily innervated by Trpm8+ terminals are calb1+, these six recordings of such neurons in Figure 4E could also be calb1+.

As noted in our initial resubmission, we fully accept that the sample size is limited. We have already toned down statements related to this, to say that our findings “strongly suggest” that the cells with dense Trpm8 input are cold-selective (both in the Abstract and Results)

**Reviewer #2 (Public review):**
In the characterization of recorded neurons in close contact or in the absence of this contact with TRPM8 afferents, the number of recorded neurons is relatively low. In addition, the strength of thermal stimuli is not very well controlled, preventing a more precise characterization of the connectivity.

The authors acknowledge that, technically, this is a very difficult preparation with very low yield as far as obtaining successful recordings. Moreover, the tissue needs to be maintained at room temperature which is obviously not ideal when characterizing cold thermoreceptors due to the unavoidable effects of low temperature on cold-activated receptors.

Please see our response to Reviewer #1 (Public review):

**Reviewer #3 (Public review):**
The main limitation remains the relatively small number of neurons that could be recorded electrophysiologically. While understandable given the complexity of the preparation, this necessarily limits generalization.

Again, please see our response to Reviewer #1 (Public review):

**Recommendations for the authors:**

**Reviewer #2 (Recommendations for the authors):**
(1) Line 609. The authors used the Trpm8Flp;RCE:FRT;Ai9 mice in some electrophysiological experiments. What is the function of the Ai9 allele (a Cre-dependent reporter) in this cross? Should not be a Cre line as well?

One of the mice used for electrophysiological experiments was Trpm8Flp;RCE:FRT;Ai9, and this animal received an injection of AAV encoding Cre into the caudal ventrolateral medulla, resulting in tdTomato expression in spinal projection neurons. This part of the Methods was inadvertently omitted from the resubmitted version (see next point). This has been corrected, and in addition, this information is shown in the cartoon in Fig 4A and is explained in the figure legend.

(2) Line 860. Phrase is incomplete

We apologise for this – 3 lines from the original version had been deleted inadvertently. This has now been corrected.

(3) Line 103 "These results are therefore consistent with the transcriptomic findings described above (36,37)."

I would revise the references used to support this claim. Reference 37 is a transcriptomic atlas of the brain. I could not find TRPM8 expression data in DRG in this reference.

Figure S4 of reference 37 deals with the mouse peripheral nervous system and describes Trpm8 classes of primary afferent. More detail on these cells (including expression of VGLUT3, Tac1, Calca and Trpv1) can be found in the associated website: mousebrain.org/adolescent/genesearch.html. We have therefore left this reference as it is.

(4) Line 242. "neurons with dense Trpm8 input had significantly lower sEPSC frequencies compared to those that lacked dense Trpm8 input".

This is an interesting paradox because cold thermoreceptors (i.e. the presumed direct monosynaptic input to these projection neurons) are known to be spontaneously active at physiological skin temperatures. This is well characterized in trigeminal corneal endings (DOI: 10.1038/nm.2264). In fact, the decrease in this spontaneous activity can be used by mice to faithfully detect warm stimuli (DOI: 10.1016/j.neuron.2020.02.035). This reviewer likes to remark that this low spontaneous frequency may be due to the non-physiological temperature of this preparations, leading to partial adaptation/desensitization of the afferents. Perhaps, it also influences the amplitude (e.g. release probability) of EPSPs (I do not expect you to do anything about my remark).

These are interesting points, but we do not feel that we can add anything here.

(5) Figure 3A. It would be useful to include orientation references (dorso-ventral, mediolateral) in the images. Same comment applies to Figure 5C.0m

Since these are horizontal sections, the axes are medio-lateral and rostro-caudal. Corresponding orientation markers have been added to both figures.

(6) Figure 3F. If I understood correctly, the light pulse used for optogenetic activation is delivered directly through the objective used for recording the cell. Thus, the distance between pre and postsynaptic neuron should be minimal. That being the case, I do not understand how a monosynaptic input can have a delay of 5 or 7 ms. Am I missing something?

The relatively long duration of latency is likely to reflect a slow rise time of depolarisation in the Trpm8 terminals, so that although channels will open very rapidly, there is a delay until the boutons reach action potential threshold. Hachisuka et al (2016) recorded from Nts^Cre;^Ai32 mice (i.e. coding for channelrhodopsin) and found typical latencies of >5 ms (Fig 5E in that paper). We believe that this delay is exacerbated by the low levels of expression of ChR2 that we were able to achieve with the neonatal i.p. injection approach. We have provided a brief explanation for this, and cited the reference in the Results section (lines 197-198).

(7) Figures 4E/H. To be meaningful, the pie charts should include the n (total number of neurons). See, for example figure 5J.

Numbers have been added to the pie charts.